# ON THE GEOMETRY OF ADVERSARIAL EXAMPLES

## ABSTRACT

Adversarial examples are a pervasive phenomenon of machine learning models where seemingly imperceptible perturbations to the input lead to misclassifications for otherwise statistically accurate models. We propose a geometric framework, drawing on tools from the manifold reconstruction literature, to analyze the high-dimensional geometry of adversarial examples. In particular, we highlight the importance of *codimension*: for low-dimensional data manifolds embedded in high-dimensional space there are many directions off the manifold in which to construct adversarial examples. Adversarial examples are a natural consequence of learning a decision boundary that classifies the low-dimensional data manifold well, but classifies points near the manifold incorrectly. Using our geometric framework we prove (1) a tradeoff between robustness under different norms, (2) that adversarial training in balls around the data is sample inefficient, and (3) sufficient sampling conditions under which nearest neighbor classifiers and ball-based adversarial training are robust.

## 1 INTRODUCTION

Deep learning at scale has led to breakthroughs on important problems in computer vision (Krizhevsky et al. (2012)), natural language processing (Wu et al. (2016)), and robotics (Levine et al. (2015)). Shortly thereafter, the interesting phenomena of *adversarial examples* was observed. A seemingly ubiquitous property of machine learning models where perturbations of the input that are imperceptible to humans reliably lead to confident incorrect classifications (Szegedy et al. (2013); Goodfellow et al. (2014)). What has ensued is a standard story from the security literature: a game of cat and mouse where defenses are proposed only to be quickly defeated by stronger attacks (Athalye et al. (2018)). This has led researchers to develop methods which are provably robust under specific attack models (Madry et al. (2018); Wong & Kolter (2018); Sinha et al. (2018); Raghunathan et al. (2018)). As machine learning proliferates into society, including security-critical settings like health care (Esteva et al. (2017)) or autonomous vehicles (Codevilla et al. (2018)), it is crucial to develop methods that allow us to understand the vulnerability of our models and design appropriate counter-measures.

In this paper, we propose a geometric framework for analyzing the phenomenon of adversarial examples. We leverage the observation that datasets encountered in practice exhibit low-dimensional structure despite being embedded in very high-dimensional input spaces. This property is colloquially referred to as the "Manifold Hypothesis": the idea that low-dimensional structure of 'real' data leads to tractable learning. We model data as being sampled from class-specific low-dimensional manifolds embedded in a high-dimensional space. We consider a threat model where an adversary may choose *any* point on the data manifold to perturb by $\epsilon$ in order to fool a classifier. In order to be robust to such an adversary, a classifier must be correct everywhere in an $\epsilon$-tube around the data manifold. Observe that, even though the data manifold is a low-dimensional object, this tube has the same dimension as the entire space the manifold is embedded in. Our analysis argues that adversarial examples are a natural consequence of learning a decision boundary that classifies all points on a low-dimensional data manifold correctly, but classifies many points near the manifold incorrectly. The high *codimension*, the difference between the dimension of the data manifold and the dimension of the embedding space, is a key source of the pervasiveness of adversarial examples.

Our paper makes the following contributions. First, we develop a geometric framework, inspired by the manifold reconstruction literature, that formalizes the manifold hypothesis described above and our attack model. Second, we highlight the role *codimension* plays in vulnerability to adversarial

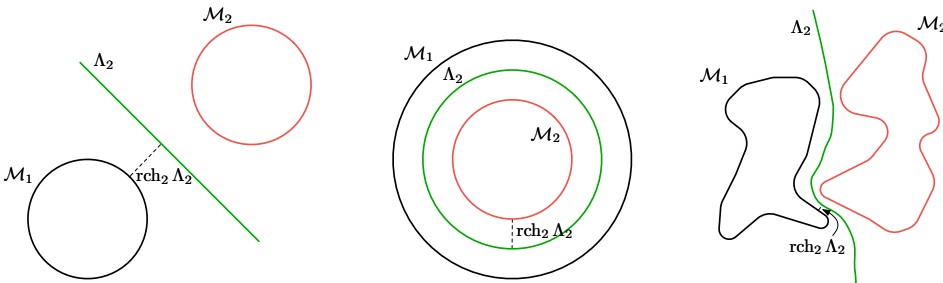

Figure 1: Examples of the decision axis $\Lambda_2$, shown here in green, for different data manifolds. Intuitively, the decision axis captures an optimal decision boundary between the data manifolds. It's optimal in the sense that each point on the decision axis is as far away from each data manifold as possible. Notice that in the first example, the decision axis coincides with the maximum margin line.

examples. As the codimension increases, there are an increasing number of directions off the data manifold in which to construct adversarial perturbations. Prior work has attributed vulnerability to adversarial examples to input dimension (Gilmer et al. (2018)). This is the first work that investigates the role of *codimension* in adversarial examples. Interestingly, we find that different classification algorithms are less sensitive to changes in codimension. Third, we apply this framework to prove the following results: (1) we show that the choice of norm to restrict an adversary is important in that there exists a tradeoff between being robust to different norms: we present a classification problem where improving robustness under the $\|\cdot\|_\infty$ norm requires a loss of $\Omega(1-1/\sqrt{d})$ in robustness to the $\|\cdot\|_2$ norm; (2) we show that a common approach, training against adversarial examples drawn from balls around the training set, is insufficient to learn robust decision boundaries with realistic amounts of data; and (3) we show that nearest neighbor classifiers do not suffer from this insufficiency, due to geometric properties of their decision boundary away from data, and thus represent a potentially robust classification algorithm. Finally we provide experimental evidence on synthetic datasets and MNIST that support our theoretical results.

## 2 RELATED WORK

This paper approaches the problem of adversarial examples using techniques and intuition from the manifold reconstruction literature. Both fields have a great deal of prior work, so we focus on only the most related papers here.

### 2.1 ADVERSARIAL EXAMPLES

Some previous work has considered the relationships between adversarial examples and high dimensional geometry. Franceschi et al. (2018) explore the robustness of classifiers to random noise in terms of distance to the decision boundary, under the assumption that the decision boundary is locally flat. The work of Gilmer et al. (2018) experimentally evaluated the setting of two concentric under-sampled 499-spheres embedded in $\mathbb{R}^{500}$, and concluded that adversarial examples occur on the data manifold. In contrast, we present a geometric framework for proving robustness guarantees for learning algorithms, that makes no assumptions on the decision boundary. We carefully sample the data manifold in order to highlight the importance of *codimension*; adversarial examples exist *even* when the manifold is perfectly classified. Additionally we explore the importance of the spacing between the constituent data manifolds and sampling requirements for learning algorithms.

Wang et al. (2018) explore the robustness of $k$-nearest neighbor classifiers to adversarial examples. In the setting where the Bayes optimal classifier is uncertain about the true label of each point, they show that $k$-nearest neighbors is not robust if $k$ is a small constant. They also show that if $k \in \Omega(\sqrt{dn \log n})$, then $k$-nearest neighbors is robust. Using our geometric framework we show a complementary result: in the setting where each point is certain of its label, 1-nearest neighbors is robust to adversarial examples.

The decision and medial axes defined in Section 3 are maximum margin decision boundaries. Hard margin SVMs define define a linear separator with maximum margin, maximum distance from the training data (Cortes & Vapnik (1995)). Kernel methods allow for maximum margin decision boundaries that are non-linear by using additional features to project the data into a higher-dimensional feature space (Shawe-Taylor & Cristianini (2004)). The decision and medial axes generalize the notion of maximum margin to account for the arbitrary curvature of the data manifolds. There have been attempts to incorporate maximum margins into deep learning (Sun et al. (2016); Liu et al. (2016); Liang et al. (2017); Elsayed et al. (2018)), often by designing loss functions that encourage large margins at either the output (Sun et al. (2016)) or at any layer (Elsayed et al. (2018)). In contrast, the decision axis is defined on the input space and we use it as an analysis tool for proving robustness guarantees.

## 2.2 MANIFOLD RECONSTRUCTION

Manifold reconstruction is the problem of discovering the structure of a $k$-dimensional manifold embedded in $\mathbb{R}^d$, given *only* a set of points sampled from the manifold. A large vein of research in manifold reconstruction develops algorithms that are *provably good*: if the points sampled from the underlying manifold are sufficiently dense, these algorithms are guaranteed to produce a geometrically accurate representation of the unknown manifold with the correct topology. The output of these algorithms is often a *simplicial complex*, a set of simplices such as triangles, tetrahedra, and higher-dimensional variants, that approximate the unknown manifold. In particular these algorithms output subsets of the Delaunay triangulation, which, along with their dual the Voronoi diagram, have properties that aid in proving geometric and topological guarantees (Edelsbrunner & Shah (1997)).

The field first focused on curve reconstruction in $\mathbb{R}^2$ (Amenta et al. (1998)) and subsequently in $\mathbb{R}^3$ (Dey & Kumar (1999)). Soon after algorithms were developed for surface reconstruction in $\mathbb{R}^3$, both in the noise-free setting (Amenta & Bern (1999); Amenta et al. (2002)) and in the presence of noise (Dey & Goswami (2004)). We borrow heavily from the analysis tools of these early works, including the medial axis and the reach. However we emphasize that we have adapted these tools to the learning setting. To the best of our knowledge, our work is the first to consider the medial axis under different norms.

In higher-dimensional embedding spaces (large $d$), manifold reconstruction algorithms face the *curse of dimensionality*. In particular, the Delaunay triangulation, which forms the bedrock of algorithms in low-dimensions, of $n$ vertices in $\mathbb{R}^d$ can have up to $\Theta(n^{\lceil d/2 \rceil})$ simplices. To circumvent the curse of dimensionality, algorithms were proposed that compute subsets of the Delaunay triangulation restricted to the $k$-dimensional tangent spaces of the manifold at each sample point (Boissonnat & Ghosh (2014)). Unfortunately, progress on higher-dimensional manifolds has been limited due to the presence of so-called "sliver" simplices, poorly shaped simplices that cause in-consistences between the local triangulations constructed in each tangent space (Cheng et al. (2005); Boissonnat & Ghosh (2014)). Techniques that provably remove sliver simplices have prohibitive sampling requirements (Cheng et al. (2000); Boissonnat & Ghosh (2014)). Even in the special case of surfaces ($k = 2$) embedded in high dimensions ($d > 3$), algorithms with practical sampling requirements have only recently been proposed (Khoury & Shewchuk (2016)). Our use of tubular neighborhoods as a tool for analysis is borrowed from Dey et al. (2005) and Khoury & Shewchuk (2016).

In this paper we are interested in *learning* robust decision boundaries, *not* reconstructing the underlying data manifolds, and so we avoid the use of Delaunay triangulations and their difficulties entirely. In Section 6 we present robustness guarantees for two learning algorithms in terms of a sampling condition on the underlying manifold. These sampling requirements scale with the dimension of the underlying manifold $k$, *not* with the dimension of the embedding space $d$.

## 3 THE GEOMETRY OF DATA

We model data as being sampled from a set of low-dimensional manifolds (with or without boundary) embedded in a high-dimensional space $\mathbb{R}^d$. We use $k$ to denote the dimension of a manifold $\mathcal{M} \subset \mathbb{R}^d$. The special case of a 1-manifold is called a *curve*, and a 2-manifold is a *surface*. The *codimension* of $\mathcal{M}$ is $d - k$, the difference between the dimension of the manifold and the dimension

of the embedding space. The "Manifold Hypothesis" is the observation that in practice, data is often sampled from manifolds, usually of high codimension.

In this paper we are primarily interested in the classification problem. Thus we model data as being sampled from $C$ *class manifolds* $\mathcal{M}_1, \ldots, \mathcal{M}_C$, one for each class. When we wish to refer to the entire space from which a dataset is sampled, we refer to the *data manifold* $\mathcal{M} = \cup_{1 \leq j \leq C} \mathcal{M}_j$. We often work with a finite sample of $n$ points, $X \subset \mathcal{M}$, and we write $X = \{X_1, X_2, \ldots, X_n\}$. Each sample point $X_i$ has an accompanying class label $y_i \in \{1, 2, \ldots, C\}$ indicating which manifold $\mathcal{M}_{y_i}$ the point $X_i$ is sampled from.

Consider a $\| \cdot \|_p$-ball $B$ centered at some point $c \in \mathbb{R}^d$ and imagine growing $B$ by increasing its radius starting from zero. For nearly all starting points $c$, the ball $B$ eventually intersects one, *and only one*, of the $\mathcal{M}_i$'s. Thus the nearest point to $c$ on $\mathcal{M}$, in the norm $\| \cdot \|_p$, lies on $\mathcal{M}_i$. (Note that the nearest point on $\mathcal{M}_i$ need not be unique.)

The *decision axis* $\Lambda_p$ of $\mathcal{M}$ is the set of points $c$ such that the boundary of $B$ intersects two or more of the $\mathcal{M}_i$, but the interior of $B$ does not intersect $\mathcal{M}$ at all. In other words, the decision axis $\Lambda_p$ is the set of points that have two or more closest points, in the norm $\| \cdot \|_p$, *on distinct class manifolds*. See Figure 1. The decision axis is inspired by the medial axis, which was first proposed by Blum (1967) in the context of image analysis and subsequently modified for the purposes of curve and surface reconstruction by Amenta et al. (1998; 2002). We have modified the definition to account for multiple class manifolds and have renamed our variant in order to avoid confusion in the future.

The decision axis $\Lambda_p$ can intuitively be thought of as a decision boundary that is optimal in the following sense. First, $\Lambda_p$ separates the class manifolds when they do not intersect (Lemma 7). Second, each point of $\Lambda_p$ is as far away from the class manifolds as possible in the norm $\| \cdot \|_p$. As shown in the leftmost example in Figure 1, in the case of two linearly separable circles of equal radius, the decision axis $\Lambda_2$ is exactly the line that separates the data with maximum margin. For arbitrary manifolds, $\Lambda_p$ generalizes the notion of maximum margin to account for the arbitrary curvature of the class manifolds.

Let $T \subset \mathbb{R}^d$ be any set. The *reach* $\mathrm{rch}_p(T; \mathcal{M})$ of $\mathcal{M}$ is defined as $\inf_{x \in \mathcal{M}, y \in T} \|x - y\|_p$. When $\mathcal{M}$ is compact, the reach is achieved by the point on $\mathcal{M}$ that is closest to $T$ under the $\| \cdot \|_p$ norm. We will drop $\mathcal{M}$ from the notation when it is understood from context.

Finally, an $\epsilon$-*tubular neighborhood* of $\mathcal{M}$ is defined as $\mathcal{M}^{\epsilon,p} = \{x \in \mathbb{R}^d : \inf_{y \in \mathcal{M}} \|x - y\|_p \leq \epsilon\}$. That is, $\mathcal{M}^{\epsilon,p}$ is the set of all points whose distance to $\mathcal{M}$ under the metric induced by $\| \cdot \|_p$ is less than $\epsilon$. Note that while $\mathcal{M}$ is $k$-dimensional, $\mathcal{M}^{\epsilon,p}$ is always $d$-dimensional. Tubular neighborhoods are how we rigorously define adversarial examples. Consider a classifier $f : \mathbb{R}^d \to [C]$ for $\mathcal{M}$. An $\epsilon$-*adversarial example* is a point $x \in \mathcal{M}_i^{\epsilon,p}$ such that $f(x) \neq i$. A classifier $f$ is robust to all $\epsilon$-adversarial examples when $f$ correctly classifies not only $\mathcal{M}$, but all of $\mathcal{M}^{\epsilon,p}$. Thus the problem of being robust to adversarial examples is rightly seen as one of *generalization*. In this paper we will be primarily concerned with exploring the conditions under which we can provably learn a decision boundary that correctly classifies $\mathcal{M}^{\epsilon,p}$. When $\epsilon < \mathrm{rch}_p \Lambda_p$, the decision axis $\Lambda_p$ is one decision boundary that correctly classifies $\mathcal{M}^{\epsilon,p}$ (Corollary 9). Throughout the remainder of the paper we will drop the $p$ in $\mathcal{M}^{\epsilon,p}$ from the notation, instead writing $\mathcal{M}^{\epsilon}$; the norm will always be clear from context.

The geometric quantities defined above can be defined more generally for any distance metric $d(\cdot, \cdot)$. In this paper we will focus exclusively on the metrics induced by the norms $\| \cdot \|_p$ for $p > 0$. The decision axis under $\| \cdot \|_2$ is in general *not* identical to the decision axis under $\| \cdot \|_\infty$. In Section 4 we will prove that since $\Lambda_2$ is not identical to $\Lambda_\infty$ there exists a tradeoff in the robustness of any decision boundary between the two norms.

## 4  A PROVABLE TRADEOFF IN ROBUSTNESS BETWEEN NORMS

Schott et al. (2018) explore the vulnerability of robust classifiers to attacks under different norms. In particular, they take the robust pretrained classifier of Madry et al. (2018), which was trained to be robust to $\| \cdot \|_\infty$-perturbations, and subject it to $\| \cdot \|_0$ and $\| \cdot \|_2$ attacks. They show that accuracy drops to $0\%$ under $\| \cdot \|_0$ attacks and to $35\%$ under $\| \cdot \|_2$. Here we explain why poor robustness under the norm $\| \cdot \|_2$ should be expected.

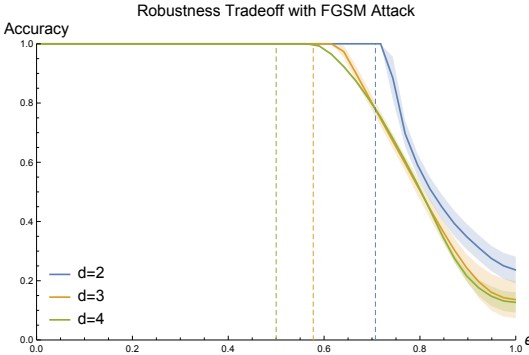

Figure 2: As the dimension increases, the $\mathrm{rch}_2\left(\Lambda_\infty; S_1 \cup S_2\right)$ decreases, and so an $\|\cdot\|_\infty$ robust classifier is less robust to $\|\cdot\|_2$ attacks. The dashed lines are placed at $1/\sqrt{d}$, where our theoretical results predict we should start finding $\|\cdot\|_2$ adversarial examples. We use the robust $\|\cdot\|_\infty$ loss of Wong & Kolter (2018)

We say a decision boundary $\mathcal{D}_f$ for a classifier $f$ is $\epsilon$-robust in the $\|\cdot\|_p$ norm if $\epsilon < \mathrm{rch}_p \mathcal{D}_f$. In words, starting from any point $x \in \mathcal{M}$, a perturbation $\eta_x$ must have $p$-norm greater than $\mathrm{rch}_p \mathcal{D}_f$ to cross the decision boundary. The most robust decision boundary to $\|\cdot\|_p$-perturbations is $\Lambda_p$. In Theorem 1 we construct a learning setting where $\Lambda_2$ is distinct from $\Lambda_\infty$. Thus, in general, *no single decision boundary can be optimally robust in all norms.*

**Theorem 1.** *Let $S_1, S_2 \subset \mathbb{R}^{d+1}$ be two concentric $d$-spheres with radii $r_1 < r_2$ respectively. Let $S = S_1 \cup S_2$ and let $\Lambda_2, \Lambda_\infty$ be the $\|\cdot\|_2$ and $\|\cdot\|_\infty$ decision axes of $S$. Then $\Lambda_2 \neq \Lambda_\infty$. Furthermore $\mathrm{rch}_2 \Lambda_\infty \in \mathcal{O}(\mathrm{rch}_2 \Lambda_2/\sqrt{d})$.*

From Theorem 1 we conclude that the minimum distance from $S_1$ to $\Lambda_\infty$ *under the* $\|\cdot\|_2$ *norm* is upper bounded as $\mathrm{rch}_2 \Lambda_\infty \in \mathcal{O}(\mathrm{rch}_2 \Lambda_2/\sqrt{d})$. If a classifier $f$ is trained to learn $\Lambda_\infty$, an adversary, starting on $S_1$, can construct an $\|\cdot\|_2$ adversarial example for a perturbation as small as $\mathcal{O}(1/\sqrt{d})$. Thus we should *expect* $f$ to be less robust to $\|\cdot\|_2$-perturbations. Figure 2 verifies this result experimentally. The proof of Theorem 1 is provided in Appendix A

We expect that $\Lambda_2 \neq \Lambda_\infty$ is the common case in practice. For example, Theorem 1 extends immediately to concentric cylinders and intertwined tori by considering 2-dimensional planar cross-sections. In general, we expect that $\Lambda_2 \neq \Lambda_\infty$ in situations where a 2-dimensional cross-section with $\mathcal{M}$ has nontrivial curvature.

Theorem 1 is important because, even in recent literature, researchers have attributed this phenomena to overfitting. Schott et al. (2018) state that "the widely recognized and by far most successful defense by Madry et al. (1) *overfits* on the $L_\infty$ metric (it's highly susceptible to $L_2$ and $L_0$ perturbations)" (emphasis ours). We disagree; the Madry et al. (2018) classifier performed exactly as intended. It learned a decision boundary that is robust under $\|\cdot\|_\infty$, which we have shown is quite different from the most robust decision boundary under $\|\cdot\|_2$.

Interestingly, the proposed models of Schott et al. (2018) also suffer from this tradeoff. Their model ABS has accuracy $80\%$ to $\|\cdot\|_2$ attacks but drops to $8\%$ for $\|\cdot\|_\infty$. Similarly their model ABS Binary has accuracy $77\%$ to $\|\cdot\|_\infty$ attacks but drops to $39\%$ for $\|\cdot\|_2$ attacks.

We reiterate, in general, no single decision boundary can be optimally robust in all norms.

## 5   $X^\epsilon$ is a Poor Model of $\mathcal{M}^\epsilon$

Madry et al. (2018) suggest training a robust classifier with the help of an adversary which, at each iteration, produces $\epsilon$-perturbations around the training set that are incorrectly classified. In our notation, this corresponds to learning a decision boundary that correctly classifies $X^\epsilon = \{x \in \mathbb{R}^d : \|x - X_i\|_2 \leq \epsilon$ for some training point $X_i\}$. We believe this approach is insufficiently robust in practice, as $X^\epsilon$ is often a poor model for $\mathcal{M}^\epsilon$. In this section, we show that the volume $\mathrm{vol}\, X^\epsilon$ is often

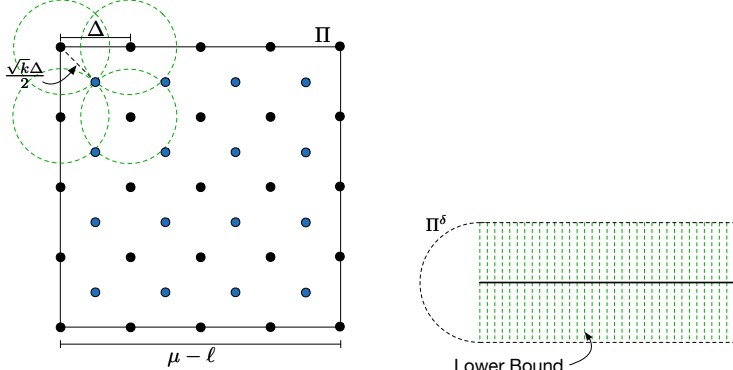

Figure 3: **Left:** To construct an $\delta$-cover we place sample points, shown here in black, along a regular grid with spacing $\Delta$. The blue points are the furthest points of $\Pi$ from the sample. To cover $\Pi$ we need $\Delta = 2\delta/\sqrt{k}$. **Right:** An illustration of the lower bound technique used in Equation 3. The volume $\text{vol}\,\Pi^\delta$ shown in the black dashed lines, is bounded from below by placing a $(d-k)$-dimensional ball of radius $\delta$ at each point of $\Pi$, shown in green. In this illustration, a 1-dimensional manifold is embedded in 2 dimensions, so these balls are 1-dimensional line segments.

a vanishingly small percentage of $\text{vol}\,\mathcal{M}^\epsilon$. These results shed light on why the ball-based learning algorithm $\mathcal{L}$ defined in Section 6 is so much less sample-efficient than nearest neighbor classifiers. In Section 7.1 we experimentally verify these observations by showing that in high-dimensional space it is easy to find adversarial examples even after training against a strong adversary. For the remainder of this section we will consider the $\|\cdot\|_2$ norm.

**Theorem 2.** *Let $\mathcal{M} \subset \mathbb{R}^d$ be a k-dimensional manifold embedded in $\mathbb{R}^d$ such that $\text{vol}_k\,\mathcal{M} < \infty$. Let $X \subset \mathcal{M}$ be a finite set of points sampled from $\mathcal{M}$. Suppose that $\epsilon \leq \text{rch}_2\,\Xi$ where $\Xi$ is the medial axis of $\mathcal{M}$, defined as in Dey (2007). Then the percentage of $\mathcal{M}^\epsilon$ covered by $X^\epsilon$ is upper bounded by*

$$\frac{\text{vol}\,X^\epsilon}{\text{vol}\,\mathcal{M}^\epsilon} \leq \frac{\pi^{k/2}\Gamma(\frac{d-k}{2}+1)}{\Gamma(\frac{d}{2}+1)}\frac{\epsilon^k}{\text{vol}_k\,\mathcal{M}}|X| \in \mathcal{O}\left(\left(\frac{2\pi}{d-k}\right)^{k/2}\frac{\epsilon^k}{\text{vol}_k\,\mathcal{M}}|X|\right). \tag{1}$$

*As the codimension $(d-k) \to \infty$, Equation 1 approaches 0, for any fixed $|X|$.*

In high codimension, even moderate under-sampling of $\mathcal{M}$ leads to a significant loss of coverage of $\mathcal{M}^\epsilon$ because the volume of the union of balls centered at the samples shrinks faster than the volume of $\mathcal{M}^\epsilon$. Theorem 2 states that in high codimensions the fraction of $\mathcal{M}^\epsilon$ covered by $X^\epsilon$ goes to 0. Almost nothing is covered by $X^\epsilon$ for training set sizes that are realistic in practice. Thus $X^\epsilon$ is a poor model of $\mathcal{M}^\epsilon$, and high classificaiton accuracy on $X^\epsilon$ does not imply high accuracy in $\mathcal{M}^\epsilon$. The proof of Theorem 2 is given in Appendix A.

Note that an alternative way of defining the ratio $\text{vol}\,X^\epsilon/\text{vol}\,\mathcal{M}^\epsilon$ is as $\text{vol}\,(X^\epsilon \cap \mathcal{M}^\epsilon)/\text{vol}\,\mathcal{M}^\epsilon$. This is equivalent in our setting since $X \subset \mathcal{M}$ and so $X^\epsilon \subset \mathcal{M}^\epsilon$.

For the remainder of the section we provide intuition for Theorem 2 by considering the special case of $k$-dimensional planes. Define $\Pi = \{x \in \mathbb{R}^d : \ell \leq x_1, \ldots, x_k \leq \mu \text{ and } x_{k+1} = \ldots = x_d = 0\}$; that is $\Pi$ is a subset of the $x_1\text{-}\ldots\text{-}x_k$-plane bounded between the coordinates $[\ell, \mu]$. A $\delta$-cover of a manifold $\mathcal{M}$ in the norm $\|\cdot\|_2$ is a finite set of points $X$ such that for every $x \in \mathcal{M}$ there exists $X_i$ such that $\|x - X_i\|_2 \leq \delta$. It is easy to construct an *explicit* $\delta$-cover $X$ of $\Pi$: place sample points at the vertices of a regular grid, shown in Figure 3 by the black vertices. The centers of the cubes of this regular grid, shown in blue in Figure 3, are the furthest points from the samples. The distance from the vertices of the grid to the centers is $\sqrt{k}\Delta/2$ where $\Delta$ is the spacing between points along an axis of the grid. To construct a $\delta$-cover we need $\sqrt{k}\Delta/2 = \delta$ which gives a spacing of $\Delta = 2\delta/\sqrt{k}$. The size of this sample is $|X| = \left(\frac{\sqrt{k}(\mu-\ell)}{2\delta}\right)^k$. Note that $|X|$ scales exponentially in $k$, the dimension of $\Pi$, not in $d$, the dimension of the embedding space.

Recall that $\Pi^\delta$ is the $\delta$-tubular neighborhood of $\Pi$. The $\delta$-balls around $X$, which comprise $X^\delta$, cover $\Pi$ and so any robust approach that guarantees correct classification within $X^\delta$ will achieve perfect accuracy on $\Pi$. However, we will show that $X^\delta$ covers only a vanishingly small fraction of $\Pi^\delta$. Let $B_\delta$ denote the $d$-ball of radius $\delta$ centered at the origin. An upper bound on the volume of $X^\delta$ is

$$\text{vol}\, X^\delta \le \text{vol}\, B_\delta |X| = \frac{\pi^{d/2}}{\Gamma(\frac{d}{2}+1)} \delta^d \left( \frac{\sqrt{k}(\mu-\ell)}{2\delta} \right)^k = \frac{\pi^{d/2}}{\Gamma(\frac{d}{2}+1)} \delta^{(d-k)} \left( \frac{\sqrt{k}(\mu-\ell)}{2} \right)^k. \quad (2)$$

Next we bound the volume $\text{vol}\, \Pi^\delta$ from below. Intuitively, a lower bound on the volume can be derived by placing a $(d-k)$-dimensional ball in the normal space at each point of $\Pi$ and integrating the volumes. Figure 3 (Right) illustrates the lower bound argument in the case of $k=1, d=2$.

$$\text{vol}\, \Pi^\delta \ge \text{vol}_{d-k}\, B_\delta^{d-k}\, \text{vol}_k\, \Pi = \frac{\pi^{(d-k)/2}}{\Gamma\left(\frac{d-k}{2}+1\right)} \delta^{d-k}(\mu-\ell)^k. \quad (3)$$

Combining Equations 2 and 3 gives an upper bound on the percentage of $\Pi^\delta$ that is covered by $X^\epsilon$.

$$\frac{\text{vol}\, X^\delta}{\text{vol}\, \Pi^\delta} \le \frac{\pi^{k/2}\Gamma\left(\frac{d-k}{2}+1\right)}{\Gamma\left(\frac{d}{2}+1\right)} \left( \frac{\sqrt{k}}{2} \right)^k. \quad (4)$$

Notice that the factors involving $\delta$ and $(\mu-\ell)$ cancel. Figure 4 (Left) shows that this expression approaches 0 as the codimension $(d-k)$ of $\Pi$ increases.

Suppose we set $\delta = 1$ and construct a 1-cover of $\Pi$. The number of points necessary to cover $\Pi$ with balls of radius 1 depends *only* on $k$, not the embedding dimension $d$. However the number of points necessary to cover the tubular neighborhood $\Pi^1$ with balls of radius 1 increases depends on *both* $k$ and $d$. In Theorem 3 we derive a lower bound on the number of samples necessary to cover $\Pi^1$.

**Theorem 3.** *Let $\Pi$ be a bounded $k$-flat as described above, bounded along each axis by $\ell < \mu$. Let $n$ denote the number of samples necessary to cover the 1-tubular neighborhood $\Pi^1$ of $\Pi$ with $\| \cdot \|_2$-balls of radius 1. That is let $n$ be the minimum value for which there exists a finite sample $X$ of size $n$ such that $\Pi^1 \subset \cup_{x \in X} B(x, 1) = X^1$. Then*

$$n \ge \frac{\pi^{-k/2}\Gamma\left(\frac{d}{2}+1\right)}{\Gamma\left(\frac{d-k}{2}+1\right)}(\mu-\ell)^k \in \Omega\left( \left( \frac{d-k}{2\pi} \right)^{k/2} (\mu-\ell)^k \right). \quad (5)$$

Theorem 3 states that, in general, it takes many fewer samples to accurately model $\mathcal{M}$ than to model $\mathcal{M}^\epsilon$. Figure 4 (Right) compares the number of points necessary to construct a 1-cover of $\Pi$ with the lower bound on the number necessary to cover $\Pi^1$ from Theorem 3. The number of points necessary to cover $\Pi^1$ increases as $\Omega\left((d-k)^{k/2}\right)$, scaling polynomially in $d$ and exponentially in $k$. In contrast, the number necessary to construct a 1-cover of $\Pi$ remains constant as $d$ increases, depending only on $k$. The proof of Theorem 3 is given in Appendix A.

Our lower bound of $\Omega\left((d-k)^{k/2}\right)$ samples is similar to the work of Schmidt et al. (2018) who prove that, in the simple Gaussian setting, robustness *requires* as much as $\Omega(\sqrt{d})$ more samples. Their arguments are statistical while ours are geometric.

Approaches that produce robust classifiers by generating adversarial examples in the $\epsilon$-balls centered on the training set do not accurately model $\mathcal{M}^\epsilon$, and it will take *many* more samples to do so. If the method behaves arbitrarily outside of the $\epsilon$-balls that define $X^\epsilon$, adversarial examples will still exist and it will likely be easy to find them. The reason deep learning has performed so well on a variety of tasks, in spite of the brittleness made apparent by adversarial examples, is because it is much easier to perform well on $\mathcal{M}$ than it is to perform well on $\mathcal{M}^\epsilon$.

## 6 PROVABLY ROBUST CLASSIFIERS

Adversarial training, the process of training on adversarial examples generated in a $\| \cdot \|_p$-ball around the training data, is a very natural approach to constructing robust models (Goodfellow et al. (2014);

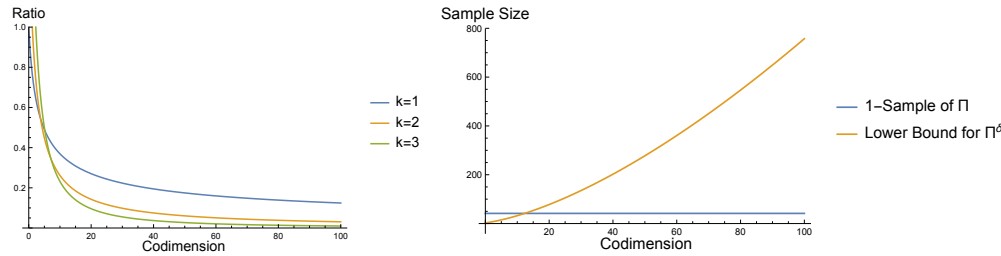

Figure 4: We plot the upper bound in Equation 4 on the left. As the codimension increases, the percentage of volume of $\Pi^1$ covered by 1-balls around the 1-sample approaches 0. On the right we plot the number of samples necessary to cover $\Pi$, shown in blue, against the number of samples necessary to cover $\Pi^1$, shown in orange, as the codimension increases.

Madry et al. (2018)). In our notation this corresponds to training on samples drawn from $X^\epsilon$ for some $\epsilon$. While natural, we show that there are simple settings where this approach is much less sample-efficient than other classification algorithms, if the *only* guarantee is correctness in $X^\epsilon$.

Define a learning algorithm $\mathcal{L}$ with the property that, given a training set $X \subset \mathcal{M}$ sampled from a manifold $\mathcal{M}$, $\mathcal{L}$ outputs a model $f_\mathcal{L}$ such that for every $x \in X$ with label $y$, and every $\hat{x} \in B(x, \mathrm{rch}_p \Lambda_p)$, $f_\mathcal{L}(\hat{x}) = f_\mathcal{L}(x) = y$. Here $B(x, r)$ denotes the ball centered at $x$ of radius $r$ in the relevant norm. That is, $\mathcal{L}$ learns a model that outputs the same label for any $\|\cdot\|_p$-perturbation of $x$ up to $\mathrm{rch}_p \Lambda_p$ as it outputs for $x$. $\mathcal{L}$ is our theoretical model of adversarial training (Goodfellow et al. (2014); Madry et al. (2018)). Theorem 4 states that $\mathcal{L}$ is sample inefficient in high codimensions.

**Theorem 4.** *There exists a classification algorithm $\mathcal{A}$ that, for a particular choice of $\mathcal{M}$, correctly classifies $\mathcal{M}^\epsilon$ using exponentially fewer samples than are required for $\mathcal{L}$ to correctly classify $\mathcal{M}^\epsilon$.*

Theorem 4 follows from Theorems 5 and 6. In Theorems 5 and 6 we will prove that a nearest neighbor classifier $f_{\mathrm{nn}}$ is one such classification algorithm. Nearest neighbor classifiers are naturally robust in high codimensions because the Voronoi cells of $X$ are *elongated in the directions normal to $\mathcal{M}$* when $X$ is dense (Dey (2007)).

Recall that a $\delta$-cover of a manifold $\mathcal{M}$ in the norm $\|\cdot\|_p$ is a finite set of points $X$ such that for every $x \in \mathcal{M}$ there exists $X_i$ such that $\|x - X_i\|_p \leq \delta$. Theorem 5 gives a sufficient sampling condition for $f_\mathcal{L}$ to correctly classify $\mathcal{M}^\epsilon$ for all manifolds $\mathcal{M}$. Theorem 5 also provides a sufficient sampling condition for a nearest neighbor classifier $f_{\mathrm{nn}}$ to correctly classify $\mathcal{M}^\epsilon$, which is substantially less dense than that of $f_\mathcal{L}$. Thus different classification algorithms have different sampling requirements in high codimensions.

**Theorem 5.** *Let $\mathcal{M} \subset \mathbb{R}^d$ be a $k$-dimensional manifold and let $\epsilon < \mathrm{rch}_p \Lambda_p$ for any $p$. Let $f_{nn}$ be a nearest neighbor classifier and let $f_\mathcal{L}$ be the output of a learning algorithm $\mathcal{L}$ as described above. Let $X_{\mathrm{nn}}, X_\mathcal{L} \subset \mathcal{M}$ denote the training sets for $f_{\mathrm{nn}}$ and $\mathcal{L}$ respectively. We have the following sampling guarantees:*

1. *If $X_{\mathrm{nn}}$ is a $\delta$-cover for $\delta \leq 2(\mathrm{rch}_p \Lambda_p - \epsilon)$ then $f_{\mathrm{nn}}$ correctly classifies $\mathcal{M}^\epsilon$.*

2. *If $X_\mathcal{L}$ is a $\delta$-cover for $\delta \leq \mathrm{rch}_p \Lambda_p - \epsilon$ then $f_\mathcal{L}$ correctly classifies $\mathcal{M}^\epsilon$.*

The bounds on $\delta$ in Theorem 5 are sufficient, but they are not always necessary. There exist manifolds where the bounds in Theorem 5 are pessimistic, and less dense samples corresponding to larger values of $\delta$ would suffice. In Theorem 6 we show a setting where bounds on $\delta$ similar to those in Theorem 5 are *necessary*. In this setting, the difference of a factor of 2 in $\delta$ between the sampling requirements of $f_{\mathrm{nn}}$ and $f_\mathcal{L}$ leads to an exponential gap between the sizes of $X_{\mathrm{nn}}$ and $X_\mathcal{L}$ necessary to achieve the same amount of robustness.

Consider two subsets of $k$-flats $\Pi_1, \Pi_2$, as defined in Section 5, where $\Pi_1$ lies in the subspace $x_d = 0$ and $\Pi_2$ lies in the subspace $x_d = 1$; thus $\mathrm{rch}_2 \Lambda_2 = 1$. In the $\|\cdot\|_2$ norm we can show that the gap in Theorem 5 is necessary for $\Pi = \Pi_1 \cup \Pi_2$. Furthermore the bounds we derive for $\delta$-covers for $\Pi$ for both $f_{\mathrm{nn}}$ and $f_\mathcal{L}$ are tight. Combined with well-known properties of covers, we get that the ratio $|X_\mathcal{L}|/|X_{\mathrm{nn}}|$ is exponential in $k$.

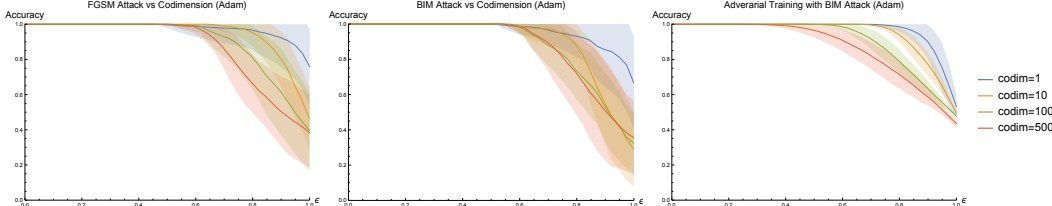

Figure 5: As the codimension increases the robustness of decision boundaries learned by Adam on naturally trained networks decreases steadily. **Left**: Effectiveness of FGSM attacks as codimension increases. **Center**: BIM attacks. **Right**: Training using the adversarial training procedure of Madry et al. (2018) is no guarantee of robustness; as the codimension increases it becomes easier to find adversarial examples using BIM attacks. Appendix B.4 shows the performance on nearest neighbor on this data, which is essentially perfect accuracy for all $\epsilon$.

**Theorem 6.** *Let $\Pi = \Pi_1 \cup \Pi_2$ as described above. Let $X_{\mathrm{nn}}, X_{\mathcal{L}} \subset \Pi$ be minimum training sets necessary to guarantee that $f_{\mathrm{nn}}$ and $f_{\mathcal{L}}$ correctly classify $\mathcal{M}^\epsilon$. Then we have that*

$$\frac{|X_{\mathcal{L}}|}{|X_{\mathrm{nn}}|} \geq 2^{k/2} \tag{6}$$

We have shown that both $\mathcal{L}$ and nearest neighbor classifiers learn robust decision boundaries when provided sufficiently dense samples of $\mathcal{M}$. However there are settings where nearest neighbors is exponentially more sample-efficient than $\mathcal{L}$ in achieving the same amount of robustness. We experimentally verify these theoretical results in Section 7.1. Proofs for all of the results in this section are provided in Appendix A.

# 7 EXPERIMENTS

## 7.1 HIGH CODIMENSION REDUCES ROBUSTNESS

Section 5 suggests that as the codimension increases it should become easier to find adversarial examples. To verify this, we introduce two synthetic datasets, CIRCLES and PLANES, which allow us to carefully vary the codimension while maintaining dense samples. The CIRCLES dataset consists of two concentric circles in the $x_1$-$x_2$-plane, with $\mathrm{rch}_2 \Lambda_2 = 1$. We densely sample 1000 random points on each circle for both the training and the test sets. The PLANES dataset consists of two 2-dimensional planes, the first in the $x_d = 0$ and the second in $x_d = 2$, so that $\mathrm{rch}_2 \Lambda_2 = 1$. We sample the training set at the vertices of the grid described in Section 5, and the test set at the centers of the grid cubes, the blue points in Figure 3. Further details are provided in Appendix E and visualizations in Appendix H.

We consider two attacks, the fast gradient sign method (FGSM) (Goodfellow et al. (2014)) and the basic iterative method (BIM) (Kurakin et al. (2016)) under $\|\cdot\|_2$. We use the implementations provided in the cleverhans library (Papernot et al. (2018)). Further implementation details are provided in Appendix E. Our experimental results are averaged over 20 retrainings of our model architecture, using Adam (Kingma & Ba (2015)). Further implementation details are provided in Appendix E. Figure 5(Left, Center) shows FGSM and BIM attacks on the CIRCLES dataset as we vary the codimension. For both attacks we see a steady decrease in robustness as we increase the codimension, on average.

Madry et al. (2018) propose training against a PGD adversary to improve robustness. Section 5 suggests that this should be insufficient to guarantee robustness, as $X^\epsilon$ is often a poor model for $\mathcal{M}^\epsilon$. We train against a PGD adversary with $\epsilon = 1$ under $\|\cdot\|_2$-perturbations on the PLANES dataset. Figure 5 (Right) shows that it is still easy to find adversarial examples for $\epsilon < 1$ and that as the codimension increases we can find adversarial examples for decreasing values of $\epsilon$. In contrast, nearest neighbor achieves perfect robustness for all epsilon on this data (see Appendix B.4 for details).

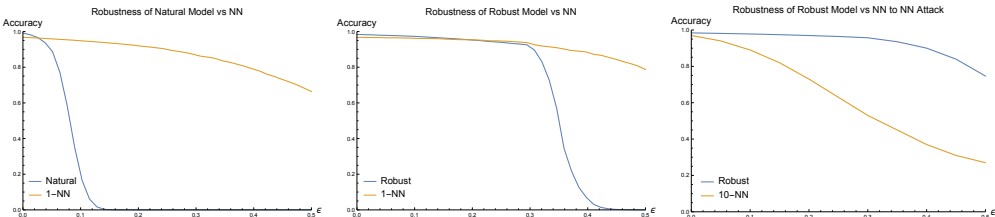

Figure 6: Robustness of nearest neighbors on MNIST. **Left:** Performance on $l_\infty$ BIM attack against a naturally trained model. **Center:** The same for the adversarially trained convolutional models of Madry et al. (2018). **Right:** Performance of the robust model and nearest neighbors on examples generated by a custom attack on nearest neighbors.

## 7.2 MNIST

To explore performance on a more realistic dataset, we compared nearest neighbors with robust and natural models on MNIST. We considered two attacks: BIM under $l_\infty$ norm against the natural and robust models as well as a custom attack against nearest neighbors. Each of these attacks are generated from the MNIST test set. Architecture details can be found in Appendix E. Figure 6 (Left) shows that nearest neighbors is substantially more robust to BIM attacks than the naturally trained model. Figure 6 (Center) shows that nearest neighbors is comparable to the robust model up to $\epsilon = 0.3$, which is the value for which the robust model was trained. After $\epsilon = 0.3$, nearest neighbors is substantially more robust to BIM attacks than the robust model. At $\epsilon = 0.5$, nearest neighbors maintains accuracy of $78\%$ to adversarial perturbations that cause the accuracy of the robust model to drop to $0\%$. In Appendix B.2 we provide a similar result for FGSM attacks.

Figure 6 (Right) shows the performance of nearest neighbors and the robust model on adversarial examples generated for nearest neighbors. The nearest neighbor attacks are generated as follows: iteratively find the $k$ nearest neighbors and compute an attack direction by walking away from the neighbors in the true class and toward the neighbors in other classes. We find that nearest neighbors is able to be tricked by this approach, but the robust model is not. This indicates that the errors of these models are distinct and suggests that ensemble methods may effectively get the best of both worlds. Additionally, a closer investigation shows strong qualitative differences between the BIM adversarial examples and the examples generated for nearest neighbors. Appendix J argues that the adversarial examples that fool nearest neighbor line up better with human intuition.

## 8 CONCLUSION

We have presented a geometric framework for proving robustness guarantees for learning algorithms. Our framework is general and can be used to describe the robustness of any classifier. We have shown that no single model can be simultaneously robust to attacks under all norms and that nearest neighbor classifiers are theoretically more sample efficient than adversarial training. Most importantly, we have highlighted the role of codimension in contributing to adversarial examples and verified our theoretical contributions with experimental results.

We believe that a geometric understanding of the decision boundaries learned by deep networks will lead to both new geometrically inspired attacks and defenses. In Appendix C we provide a novel gradient-free geometric attack in support of this claim. Finally we believe future work into the geometric properties of decision boundaries learned by various optimization procedures will provide new techniques for black-box attacks.

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

## A    OMITTED PROOFS

### A.1    AUXILIARY LEMMAS

**Lemma 7.** *Let $\mathcal{M}_1, \mathcal{M}_2 \subset \mathbb{R}^d$ be $k$-dimensional manifolds such that $\mathcal{M} \cap \mathcal{M}_2 = \emptyset$. Let $\Lambda_p$ be their decision axis for any $p$ and let $\gamma : [0,1] \to \mathbb{R}^d$ be any path such that $\gamma(0) \in \mathcal{M}_1$ and $\gamma(1) \in \mathcal{M}_2$. Then $\gamma \cap M \neq \emptyset$, that is $\gamma$ must cross the decision axis.*

*Proof.* Define $f_1, f_2 : [0,1] \to \mathbb{R}$ as $f_1(t) = d(\gamma(t), \mathcal{M}_1)$ and $f_2(t) = d(\gamma(t), \mathcal{M}_2)$. Consider the function $g(t) = f_1(t) - f_2(t)$. Since $\mathcal{M}_1 \cap \mathcal{M}_2 = \emptyset$ and $\gamma$ starts on $\mathcal{M}_1$ and terminates on $\mathcal{M}_2$ the function $g(0) < 0$ and $g(1) > 0$. Then, since $g$ is continuous, the Intermediate Value Theorem implies that there exists $t_1 \in [0,1]$ such that $g(t_1) = 0$. Thus $d(\gamma(t_1), \mathcal{M}_1) = d(\gamma(t_1), \mathcal{M}_2)$, which implies that $\gamma(t_1)$ is on the decision axis $\Lambda$.  $\square$

**Theorem 8.** *Let $f$ be any classifier on $\mathcal{M} = \mathcal{M}_1 \cup \mathcal{M}_2$. The maximum accuracy achievable, assuming a uniform distribution, on $\mathcal{M}^\epsilon$ is*

$$1 - \frac{1}{2} \frac{\mathrm{vol}(\mathcal{M}_1^\epsilon \cap \mathcal{M}_2^\epsilon)}{\mathrm{vol}(\mathcal{M}_1^\epsilon \cup \mathcal{M}_2^\epsilon)}. \tag{7}$$

*Proof.* It is clearly optimal to classify points in $\mathrm{vol}(\mathcal{M}_1^\epsilon \setminus \mathcal{M}_2^\epsilon)$ as class 1 and to classify points in $\mathrm{vol}(\mathcal{M}_2^\epsilon \setminus \mathcal{M}_1^\epsilon)$ as class 2. Such a classifier can only be wrong when points lie in this intersection. For points in this intersection, the probability of a misclassification is $\frac{1}{2}$ for any classification that $f$ makes. Thus, the probability of misclassification is

$$\frac{1}{2} \frac{\mathrm{vol}(\mathcal{M}_1^\epsilon \cap \mathcal{M}_2^\epsilon)}{\mathrm{vol}(\mathcal{M}_1^\epsilon \cup \mathcal{M}_2^\epsilon)}.$$

$\square$

**Corollary 9.** *For $\epsilon < \mathrm{rch}_p(\Lambda_p; \mathcal{M})$ there exists a decision boundary that correctly classifies $\mathcal{M}^\epsilon$.*

*Proof.* For $\epsilon < \mathrm{rch}_p \Lambda_p$, $\mathcal{M}^\epsilon \cap \Lambda_p = \emptyset$ and so $\Lambda_p$ is one such decision boundary.  $\square$

### A.2    PROOF OF THEOREM 1

*Proof.* The decision axis under $\| \cdot \|_2$, $\Lambda_2$, is just the $d$-sphere with radius $(r_1 + r_2)/2$. However, $\Lambda_\infty$ is *not* identical to $\Lambda_2$ in this setting; in fact most $\Lambda_\infty$ of approaches $S_1$ as $d$ increases.

The geometry of a $\| \cdot \|_\infty$-ball $B_\Delta$ centered at $m \in \mathbb{R}^d$ with radius $\Delta$ is that of a hypercube centered at $m$ with side length $2\Delta$. To find a point on $\Lambda_\infty$ we place $B_\Delta$ tangent to the north pole $q$ of $S_1$ so that the corners of $B_\Delta$ touch $S_2$. The north pole has coordinate representation $q = (0, \ldots, 0, r_1)$, the

center $m = (0, \dots, 0, r_1 + \Delta)$, and a corner of $B_\Delta$ can be expressed as $p = (\Delta, \dots, \Delta, r_1 + 2\Delta)$. Additionally we have the constraint that $\|p\|_2 = r_2$ since $p \in S_2$. Then we can solve for $\Delta$ as

$$r_2^2 = \|p\|_2^2 = (d-1)\Delta^2 + (r_1 + 2\Delta)^2 = (d+3)\Delta^2 + 4r_1\Delta + r_1^2;$$

$$\Delta = \frac{-2r_1 + \sqrt{r_1^2 + 3r_2^2 + d(r_2^2 - r_1^2)}}{d+3},$$

where the last step follows from the quadratic formula and the fact that $\Delta > 0$. For fixed $r_1, r_2$, the value $\Delta$ scales as $\mathcal{O}(1/\sqrt{d})$. It follows that $\mathrm{rch}_2 \, \Lambda_\infty \in \mathcal{O}(\mathrm{rch}_2 \, \Lambda_2/\sqrt{d})$. $\qquad\square$

### A.3 Proof of Theorem 2

*Proof.* Assuming the balls centered on the samples in $X$ are disjoint we get the upper bound

$$\mathrm{vol}\, X^\epsilon \le \mathrm{vol}\, B_\epsilon |X| = \frac{\pi^{d/2}}{\Gamma(\frac{d}{2}+1)} \epsilon^d |X|. \tag{8}$$

This is identical to the reasoning in Equation 2.

The medial axis $\Xi$ of $\mathcal{M}$ is defined as the closure of the set of all points in $\mathbb{R}^d$ that have two or more closest points on $\mathcal{M}$ in the norm $\|\cdot\|_2$. The medial axis $\Xi$ is similar to the decision axis $\Lambda_2$, except that the nearest points do not need to be on distinct class manifolds. For $\epsilon \le \mathrm{rch}_2 \, \Xi$, we have the lower bound

$$\mathrm{vol}\, \mathcal{M}^\epsilon \ge \mathrm{vol}_{d-k}\, B_\epsilon^{d-k} \, \mathrm{vol}_k \, \mathcal{M} = \frac{\pi^{(d-k)/2}}{\Gamma\left(\frac{d-k}{2}+1\right)} \epsilon^{d-k} \, \mathrm{vol}_k \, \mathcal{M}. \tag{9}$$

Combining Equations 8 and 9 gives the result. To get the asymptotic result we apply Stirling's approximation to get

$$\frac{\Gamma(\frac{d-k}{2}+1)}{\Gamma(\frac{d}{2}+1)} \approx (2e)^{k/2} \frac{(d-k)^{(d-k+1)/2}}{d^{(d+1)/2}}$$

$$= (2e)^{k/2} \frac{\left(\frac{d-k}{d}\right)^{(d+1)/2}}{(d-k)^{k/2}}$$

$$= (2e)^{k/2} \frac{\left(1 - \frac{k}{d}\right)^{(d+1)/2}}{(d-k)^{k/2}}$$

$$\approx \left(\frac{2}{d-k}\right)^{k/2}.$$

The last step follows from the fact that $\lim_{d\to\infty}(1 - k/d)^{(d+1)/2} = e^{-k/2}$, where $e$ is the base of the natural logarithm. $\qquad\square$

### A.4 Proof of Theorem 3

*Proof.* We first construct an upper bound by generously assuming that the balls centered at the samples are disjoint. That is

$$\frac{\mathrm{vol}\, X^\delta}{\mathrm{vol}\, \Pi^\delta} \le \frac{n \, \mathrm{vol}\, B_\delta}{\mathrm{vol}\, \Pi^\delta}. \tag{10}$$

To guarantee that $\Pi^1 \subset \cup_{x\in X} B(x,1) = X^1$ we set the left hand side of Equation 10 equal to 1 and solve for $n$.

$$1 = \frac{\mathrm{vol}\, X^\delta}{\mathrm{vol}\, \Pi^\delta} \le \frac{n \, \mathrm{vol}\, B_\delta}{\mathrm{vol}\, \Pi^\delta}$$

$$n \ge \frac{\mathrm{vol}\, \Pi^\delta}{\mathrm{vol}\, B_\delta}$$

$$\ge \frac{\pi^{-k/2}\Gamma\left(\frac{d}{2}+1\right)}{\Gamma\left(\frac{d-k}{2}+1\right)} \left(\frac{\mu-\ell}{\delta}\right)^k$$

The last inequality follows from Equation 3. Setting $\delta = 1$ gives the result. The asymptotic result is similar to the argument in the proof of Theorem 2. $\qquad\square$

### A.5 PROOF OF THEOREM 5

*Proof.* We begin by proving (1). Let $q \in \mathcal{M}^\epsilon$ be any point in $\mathcal{M}^\epsilon$. Suppose without loss of generality that $q \in \mathcal{M}_i^\epsilon$ for some class $i$. The distance $d(q, \mathcal{M}_j)$ from $q$ to any other data manifold $\mathcal{M}_j$, and thus any sample on $\mathcal{M}_j$, is lower bounded by $d(q, \mathcal{M}_j) \geq 2 \operatorname{rch}_p \Lambda_p - \epsilon$. It is then both necessary and sufficient that there exists a $x \in \mathcal{M}_i$ such that $d(q, x) < 2 \operatorname{rch}_p \Lambda_p - \epsilon$ for $f_{\mathrm{nn}}(q) = i$. (Necessary since a properly placed sample on $\mathcal{M}_j$ can achieve the lower bound on $d(q, \mathcal{M}_j)$.) The distance from $q$ to the nearest sample $x$ on $\mathcal{M}_i$ is $d(q, x) \leq \epsilon + \delta$ for some $\delta > 0$. The question is how large can we allow $\delta$ to be and still guarantee that $f_{\mathrm{nn}}$ correctly classifies $\mathcal{M}^\epsilon$? We need

$$d(q, x) \leq \epsilon + \delta \leq 2 \operatorname{rch}_p \Lambda_p - \epsilon \leq d(q, \mathcal{M}_j)$$

which implies that $\delta \leq 2(\operatorname{rch}_p \Lambda_p - \epsilon)$. It follows that a $\delta$-cover with $\delta = 2(\operatorname{rch}_p \Lambda_p - \epsilon)$ is sufficient, and in some cases necessary, to guarantee that $f_{nn}$ correctly classifies $\mathcal{M}^\epsilon$.

Next we prove (2). As before let $q \in \mathcal{M}_i^\epsilon$. It is both necessary and sufficient for $q \in B_{\operatorname{rch}_p \Lambda_p}(x)$ for some sample $x \in \mathcal{M}_i$ to guarantee that $f_{\mathcal{L}}(q) = i$, by definition of $\mathcal{L}$. The distance to the nearest sample $x$ on $\mathcal{M}_i$ is $d(q, x) \leq \epsilon + \delta$ for some $\delta > 0$. Thus it suffices that $\delta \leq \operatorname{rch}_p \Lambda_p - \epsilon$. $\qquad\square$

### A.6 PROOF OF THEOREM 6

*Proof.* Let $q \in \Pi_1^\epsilon$. Since $\Pi_1$ is flat, the distance to from $q$ to the nearest sample $x \in \Pi_1$ is bounded as $\|q - x\|_2 \leq \sqrt{\epsilon^2 + \delta^2}$. For $f_{\mathrm{nn}}(q) = 1$ we need that $\|q - x\|_2 \leq 2 - \epsilon$, and so it suffices that $\delta \leq 2\sqrt{1 - \epsilon}$. In this setting, this is also necessary; should $\delta$ be any larger a property placed sample on $\Pi_2$ can claim $q$ in its Voronoi cell.

Similarly for $f_{\mathcal{L}}(q) = 1$ we need that $\|q - x\|_2 \leq 1$, and so it suffices that $\delta \leq \sqrt{1 - \epsilon^2}$. In this setting, this is also necessary; should $\delta$ be any larger, $q$ lies outside of every $\|\cdot\|_2$-ball $B_1(x)$ and so $\mathcal{L}$ is free to learn a decision boundary that misclassifies $q$.

Let $\mathcal{N}(\delta, \mathcal{M})$ denote the size of the minimum $\delta$-cover of $\mathcal{M}$. Since $\Pi$ is flat (has no curvature) and since the intersection of $\Pi$ with a $d$-ball centered at a point on $\Pi$ is a $k$-ball, a standard volume argument can be applied in the affine subspace aff $\Pi$ to conclude that $\mathcal{N}(\delta, \Pi) \in \Theta\left(\operatorname{vol}_k \Pi / \delta^k\right)$. So we have

$$\frac{\mathcal{N}(\sqrt{1 - \epsilon^2}, \Pi)}{\mathcal{N}(2\sqrt{1 - \epsilon}, \Pi)} = 2^k \left(\frac{1}{1 + \epsilon}\right)^{k/2}$$
$$\geq 2^{k/2}$$

Since $\Pi$ is constant in both settings, the factor $\operatorname{vol}_k \Pi$ as well as the constant factors hidden by $\Theta(\cdot)$ cancel. (Note that we are using the fact that $\Pi_1, \Pi_2$ have finite $k$-dimensional volume.) The inequality follows from the fact that the expression $(1 + \epsilon)^{-k/2}$ is monotonically decreasing on the interval $[0, 1]$ and takes value $2^{-k/2}$ at $\epsilon = 1$. $\qquad\square$

## B ADDITIONAL EXPERIMENTS

We present additional experiments to support our theoretical predictions. We reproduce the results of Section 7 using different optimization algorithms (Section B.1) and attack methods (Section B.2). These additional experiments are consistent with our conclusions in Section 7. Additionally we provide evidence that adversarial perturbations lie mostly in the directions of the normal space (Section B.3). We show that a nearest neighbor classifier is robust in high codimensions (Section B.4). Finally we show that increasing the sampling density substantially does not notably improve the robustness of adversarial training (Section B.5).

## B.1 REPRODUCING RESULTS USING SGD

In Section 7.1 we showed that increasing the codimension reduces the robustness of the decision boundaries learned by Adam on CIRCLES. In Figure 7 we reproduce this result using SGD. Again we see that as we increase the codimension the robustness decreases. SGD presents with much less variances than Adam, which we attribute to implicit regularization that has been observed for SGD (Soudry et al. (2018))

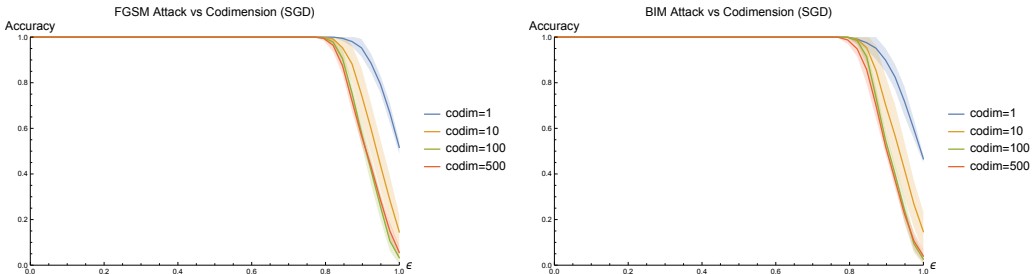

Figure 7: As in the case of training with Adam, we see a steady decrease in robustness on the CIRCLES dataset as the codimension increases when training with SGD.

Next we consider the adversarial training procedure of Madry et al. (2018) using SGD instead of Adam. We note that the authors of Madry et al. (2018) use Adam in their own experiments. Figure 8 shows that the result is consist with the result in Section 7.1. Again SGD presents with lower variance.

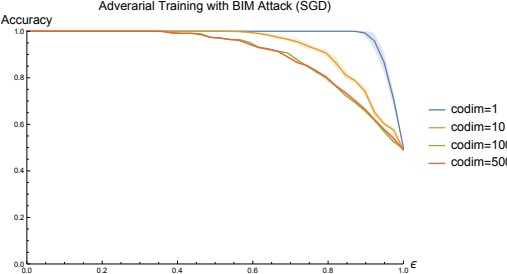

Figure 8: Adverarial training with a PGD adversary, as in Figure 5, using SGD. Similarly we see a drop in robustness as the codimension increases.

## B.2 REPRODUCING RESULTS USING FGSM

In Section 7.1 we evaluated the robustness of nearest neighbors against BIM attacks under the $\|\cdot\|_\infty$ on MNIST. In Figure 9 we evaluate the robustness of nearest neighbors against FGSM attacks under the $\|\cdot\|_\infty$ on MNIST. We use the naturally pretrained (natural) and adversarially pretrained (robust) convolutional models provided by Madry et al. (2018)[1]. Figure 9 (Left) shows that nearest neighbors is substantially more robust to FGSM attacks than the naturally trained model. Figure 9 (Right) shows that nearest neighbors is comparable to the robust model up to $\epsilon = 0.3$, which is the value for which the robust model was trained. After $\epsilon = 0.3$, nearest neighbors is substantially more robust to FGSM attacks than the robust model. At $\epsilon = 0.5$, nearest neighbors maintains accuracy of 78% to adversarial perturbations that cause the accuracy of the robust model to drop to 39%.

---

[1] https://github.com/MadryLab/mnist_challenge

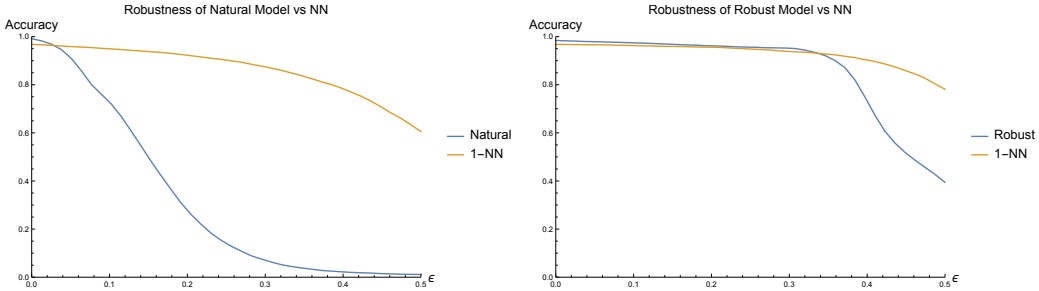

Figure 9: Robustness of nearest neighbors against the naturally trained (left) and adversarially trained (right) convolutional models of Madry et al. (2018) against FGSM attacks under the $\|\cdot\|_\infty$ norm on MNIST.

### B.3 ADVERSARIAL PERTURBATIONS ARE IN THE DIRECTIONS NORMAL TO THE DATA MANIFOLD

Let $\eta_x$ be an adversarial perturbation generated by FGSM with $\epsilon = 1$ at $x \in \mathcal{M}$. Note that the adversarial example is constructed as $\hat{x} = x + \eta_x$. In Figure 10 we plot a histogram of the angles $\angle(\eta_x, N_x\mathcal{M})$ between $\eta_x$ and the normal space $N_x\mathcal{M}$ for the CIRCLES dataset in codimensions $1, 10, 100,$ and $500$. In codimension 1, $88\%$ of adversarial perturbations make an angle of less than $10°$ with the normal space. Similarly in codimension 10, $97\%$, in codimension 100, $96\%$, and in codimension 500, $93\%$. As Figure 10 shows, nearly all adversarial perturbations make an angle less than $20°$ with the normal space. Our results are averaged over 20 retrainings of the model using SGD.

Throughout this paper we've argued that high codimension is a key source of the pervasiveness of adversarial examples. Figure 10 shows that adversarial perturbations are well aligned with the normal space. When the codimension is high, there are many directions normal to the manifold and thus many directions in which to construct adversarial perturbations.

### B.4 NEAREST NEIGHBORS IS ROBUST IN HIGH CODIMENSION

In Section 7.1 we showed that the robustness of learned decision boundaries decreased as the codimension increased. In Figure 11 we repeat the experiment in Figure 5, in which we measured the robustnesss of our neural network models to FGSM attacks as the codimension increased. We repeat this experiment using nearest neighbors to classify the adversarial examples generated by FGSM. Figure 11 shows that nearest neighbors is robust even when the codimension is high, as long as the low-dimensional data manifold is well sampled. This is a consquence of the fact that the Voronoi cells of the samples are elongated in the directions normal to the data manifold when the sample is dense.

### B.5 SAMPLING MORE DENSELY

The PLANES dataset is sampled so that the trianing set is a 1-cover of the underlying planes, which requires 450 sample points. Figure 13 shows the results of increasing the sampling density to a 0.5-cover (1682 samples) and a 0.25-cover (6498 samples). Increasing the sampling density improves the robustness of adversarial training at the same codimension and particularly in low-codimension. However adversarial training with a substantially larger training set does not produce a classifier as robust as a nearest neighbor classifier on a much smaller training set. Nearest neighbors is much more sample efficient than adversarial training, as predicted by Theorem 5 and experimentally verified in Section B.4.

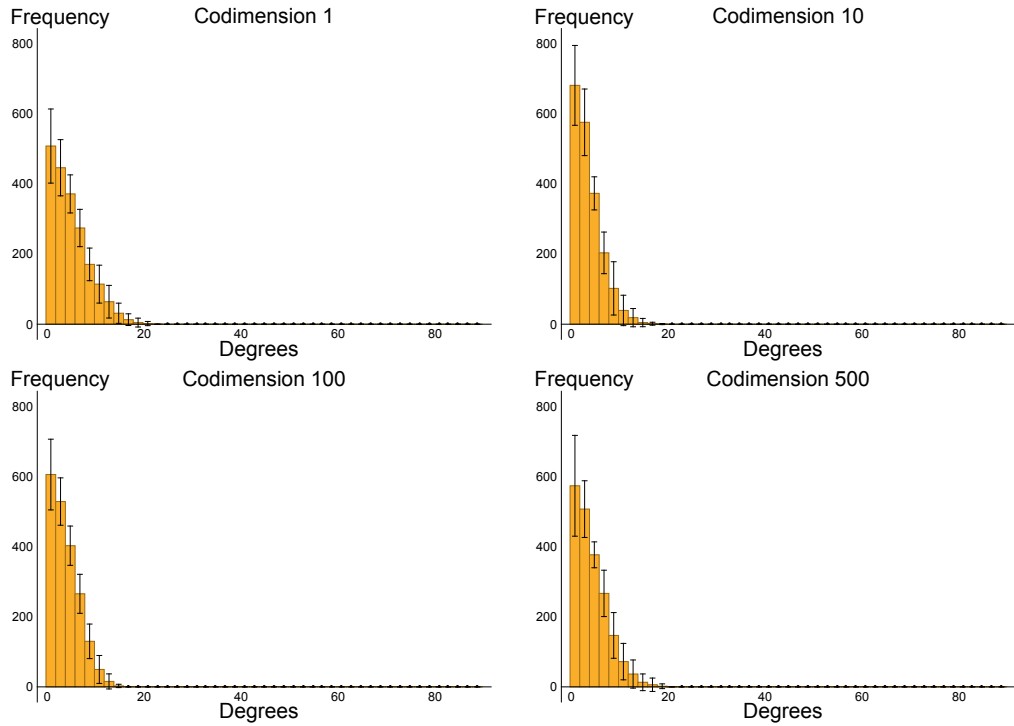

Figure 10: Histograms of the angle deviations of FGSM perturbations from the normal space for the CIRCLES dataset in codimensions 1 (upper right), 10 (upper left), 100 (lower left), 500 (lower right). Nearly all perturbations make an angle of less than $20°$ with the normal space.

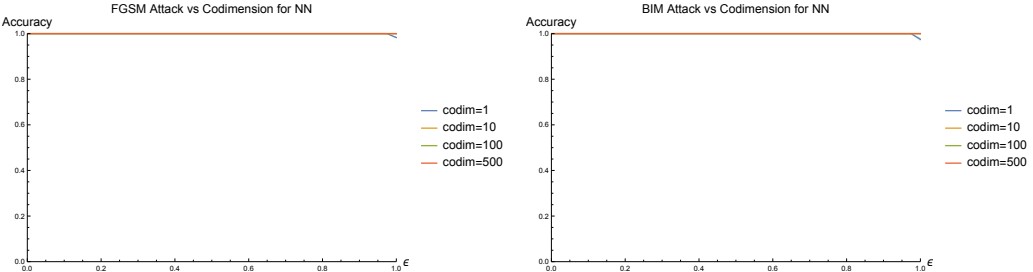

Figure 11: The FGSM (left) and BIM (right) perturbations that fooled our deep networks are correctly classified by a nearest neighbor classifier. Nearest neighbor classifiers are robust in high codimension settings because their decision boundaries are elongated in the directions normal to the data manifold.

## C  A GRADIENT-FREE GEOMETRIC ATTACK

Most current attacks rely on the gradient of the loss function at a test sample to find a direction towards the decision boundary. Partial resistance against such attacks can be achieved by obfuscating the gradients, but Athalye et al. (2018) showed how to circumvent such defenses. Brendel et al. (2018) propose a gradient-free attack for $\|\cdot\|_2$, that starts from a misclassifed point and walks toward the original point.

In this section we propose a gradient-free attack that only requires oracle access to a model, meaning we only query the model for a prediction. Consider a point $x \in X_{\text{test}}$ and the $\|\cdot\|_p$-ball $B_r(x)$ centered at $x$ of radius $r$. To construct an adversarial perturbation $\eta_x \in B_r(x)$, giving an adversarial example $\hat{x} = x + \eta_x$, we project every point in $X_{\text{test}}$ onto $B_r(x)$ and query the oracle for a prediction

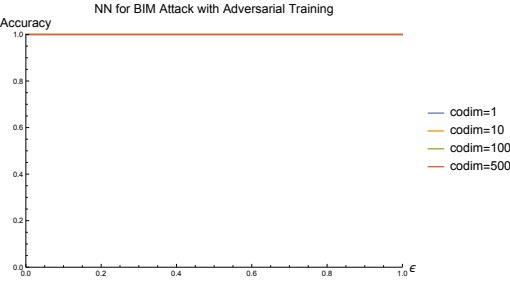

Figure 12: The BIM perturbations that fooled the adversarially trained model using the procedure suggested by Madry et al. (2018) are correctly classied by a nearest neighbor classifier.

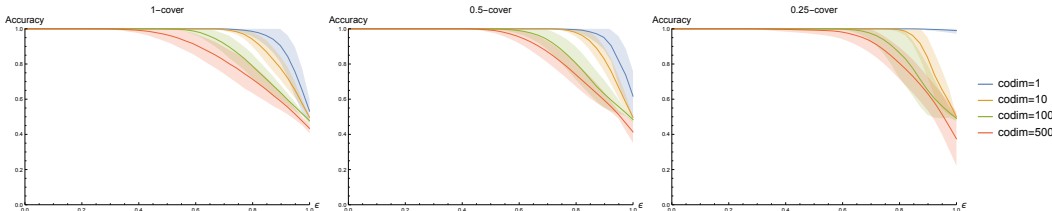

Figure 13: Adversarial training of Madry et al. (2018) on the PLANES dataset with a 1-cover (left), consisting of 450 samples, a 0.5-cover (center), 1682 samples, and a 0.25-cover (right), 6498 samples. Increasing the sampling density improves robustness at the same codimension. However even training on a significantly denser training set does not produce a classifier as robust as a nearest neighbor classifier on a much sparser training set, Figure 12

for each point. If $y \in X_{\text{test}}$ projected to a point $y'$ that the model classified differently than $x$, we take $\eta_x = y' - x$, otherwise $\eta_x = 0$. This incredibly simple attack reduces the accuracy of the pretrained robust model of Madry et al. (2018) for $\| \cdot \|_{\infty}$ and $\epsilon = 0.3$ to 90.6%, less than two percent shy of the current SOTA for whitebox attacks, 88.79% (Zheng et al. (2018)).

Simple datasets, such as CIRCLES and PLANES, allow us to diagnose issues in learning algorithms in settings where we understand how the algorithm should behave. For example Athalye et al. (2018) state that the work of Madry et al. (2018) does not suffer from obfuscated gradients. In Appendix D we provide evidence that Madry et al. (2018) *does* suffer from the obfuscated gradients problem, failing one of Athalye et al. (2018)'s criteria for detecting obfuscated gradients.

## D   THE MADRY DEFENSE SUFFERS FROM OBFUSCATED GRADIENTS

Athalye et al. (2018) identified the problem of "obfuscated gradients", a type of a gradient masking (Papernot et al. (2017)) that many proposed defenses employed to defend against adversarial examples. They identified three different types of obfuscated gradients: shattered gradients, stochastic gradients, and exploding/vanishing gradients. They examined nine recently proposed defenses, concluded that seven suffered from at least one type of obfuscated gradient, and showed how to circumvent each type of obfuscated gradient and thus each defense that employed obfuscated gradients.

Regarding the work of Madry et al. (2018), Athalye et al. (2018) stated "We believe this approach does not cause obfuscated gradients". They note that "our experiments with optimization based attacks do succeed with some probability". In this section we provide evidence that the defense of Madry et al. (2018) *does* suffer from obfuscated gradients, specifically shattered gradients. Shattered gradients occur when a defence causes the gradient field to be "nonexistent or incorrect" (Athalye et al. (2018)). Specifically we provide evidence that the defense of Madry et al. (2018) works by shattering the gradient field of the loss function around the data manifolds.

In Figure 14 (Left) we show the normalized gradient field of the loss function for a network trained on a 2-dimensional version of our PLANES dataset using the adversarial training procedure of Madry et al. (2018) with a PGD adversary. While the gradients have meaningful directions, Figure 14 (Left) shows that magnitude of the gradient field is nearly 0 everywhere around the data manifolds, which are at $y = 0$ and $y = 2$. The only notable gradients are near the decision axis which is at $y = 1$.

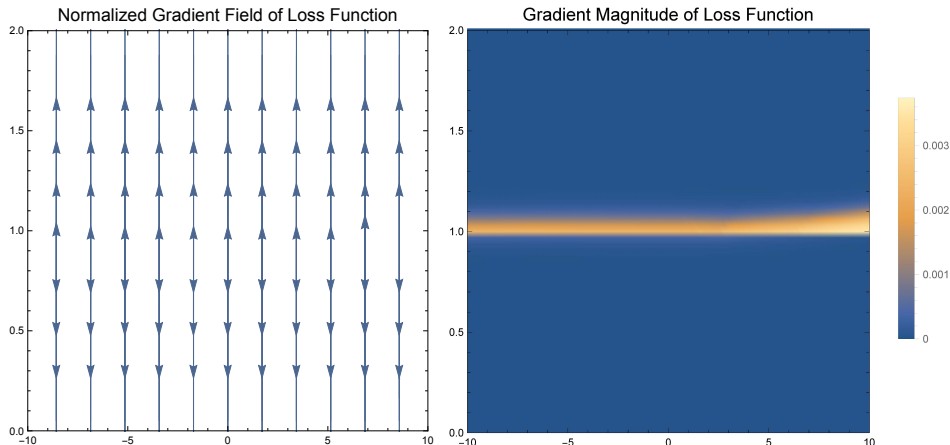

Figure 14: (Left) The normalized gradient field of the loss for an adversarially trained network. (Right) The magnitudes of the gradient. Notice that the gradients are largely 0 except at the decision axis $y = 1$.

One criteria that Athalye et al. (2018) propose for identifying obfuscated gradients is whether one-step attacks perform better than iterative attacks. The reason this criteria is useful for identifying obfuscated gradients is because one-step attacks like FGSM first normalize the gradient, ignoring its magnitude, then take as large of a step as allowed in the direction of the normalized gradient. So long as the gradient *on the manifold* points towards the decision boundary, FGSM will be effective at finding an adversarial example.

In Figure 15 we show the adversarial examples generated using PGD (left), FGSM (center), and BIM (right) for $\epsilon = 1$ starting at the test set for the PLANES dataset. FGSM produces adversarial examples at the decision axis $y = 1$, exactly where we would expect. Notice that all of the adversarial perturbation is normal to the data manifold, suggesting that the gradient on the manifold points towards the decision boundary. However the adversarial examples produced by PGD lie closer to the manifold from which the example was generated.

PGD splits the total perturbation between both the normal and the tangent spaces of the data manifold, as shown by the arrows in Figure 15. This suggests that, when trained adversarially, the network learned a gradient field that has small but correct gradients on the data manifold, but gradients that curve in the tangent directions immediately *off the manifold*.

Lastly notice that BIM, another iterative method, also produces adversarial examples that are near the decision axis. Athalye et al. (2018) cite success with iterative based optimization procedures as evidence against obfuscated gradients. However BIM also ignores the magnitude of the gradient, as it simply applies FGSM iteratively. The network has learned a gradient field that is overfit to the particulars of the PGD attack. BIM successfully navigates this gradient field, while PGD does not. While the network is robust to PGD attacks at test time, it is less robust to FGSM and BIM attacks.

## E   IMPLEMENTATION DETAILS

In Section 7 we introduced two synthetic datasets, CIRCLES and PLANES. The CIRCLES dataset consists of two concentric circles, the first with radius $r_1 = 1$ and the second with radius $r_2 = 3$, so that the rch $= 1$. The PLANES dataset consists of two 2-dimensional planes, the first in the subspace defined by $x_d = 0$, and the second in $x_d = 2$, so that rch $= 1$. The first two axis of both planes are

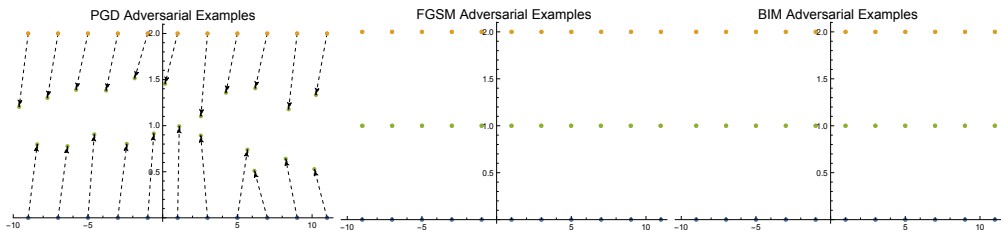

Figure 15: Adverarial examples generated using PGD (left), FGSM (center), and BIM (right). While the network is robust to PGD attacks, FGSM and BIM attacks are more effective because they ignore the magnitude of the gradient. For PGD we draw arrows from the test sample to the adversarial example generated from that point to aid the reader.

bounded as $-10 \leq x_1, x_2 \leq 10$, while $x_3 = \ldots = x_{d-1} = 0$. Both planes are sampled as described in Section 5, so that $X^1$ covers the underlying planes, where $X$ is the training set.

We consider three attacks, FGSM, BIM, and PGD, primarily under the $\|\cdot\|_2$ norm. For the iterative attacks BIM and PGD, we set the number of iterations to 30 with a step size of $\epsilon_{\text{step}} = 0.05$ per iteration.

Our controlled experiments on synthetic data consider a fully connected network with 1 hidden layer, 100 hidden units, and ReLU activations. This model architecture is more than capable of representing a nearly perfect robust decision boundary for both CIRCLES and PLANES, the latter of which is linearly separable. We set the learning rate for Adam as $\alpha = 0.1$, which we found to work best for our datasets. The parameters for the exponential decay of the first and second moment estimates were set to $\beta_1 = 0.9$ and $\beta_2 = 0.999$. We set the learning rate for SGD as $\alpha = 0.1$ and decrease the learning rate by a factor of 10 every 100 epochs. We train all of our models for 250 epochs, following Wilson et al. (2017).

All of our experiments are implemented using PyTorch. When comparing against a published result we use publicly available repositories, if able. For the robust loss of Wong & Kolter (2018), we use the code provided by the authors[2].The provided implementation[3] of the adversarial training procedure of Madry et al. (2018) considers a PGD adversary with $\|\cdot\|_\infty$-perturbations. We reimplemented their adversarial training procedure for $\|\cdot\|_2$-perturbations following their implementation and using the PGD attack implemented in the cleverhans library (Papernot et al. (2018)).

The models of Madry et al. (2018) consist of two convolutional layers with 32 and 64 filters respectively, each followed by $2 \times 2$ max pooling. After the two convolutional layers, there are two fully connected layers each with 1024 hidden units.

## F    VOLUME ARGUMENTS FOR $d$-SPHERES

Let $S \subset \mathbb{R}^{d+1}$ be a unit $d$-sphere embedded in $\mathbb{R}^{d+1}$. The volume of $S^\epsilon$ is given by

$$\text{vol } S^\epsilon = \frac{\pi^{d/2}((1+\epsilon)^d - (1-\epsilon)^d)}{\Gamma(1 + \frac{d}{2})}, \tag{11}$$

where $\Gamma$ denotes the gamma function. Let $X \subset S$ be a finite sample of size $n$ of $S$. The set $X^\epsilon$ is the set of all $\epsilon$ perturbations of points in $X$ under the norm $\|\cdot\|_2$. How well does $X^\epsilon$ approximate $S^\epsilon$ as a function of $n, d$ and $\epsilon$?

To answer this question we upper bound the ratio $\text{vol } X^\epsilon / \text{vol } S^\epsilon$ by generously assuming that the balls $B(X_i, \epsilon)$ are disjoint. The resulting upper bound is

$$\frac{\text{vol } X^\epsilon}{\text{vol } S^\epsilon} \leq \frac{n \text{ vol } B_\epsilon}{\text{vol } S^\epsilon} = \frac{n\epsilon^d}{(1+\epsilon)^d - (1-\epsilon)^d}. \tag{12}$$

---

[2]https://github.com/locuslab/convex_adversarial
[3]https://github.com/MadryLab/mnist_challenge

In Figure 16 we show three different views of this bound. In Figure 16 (Left) we set $n = 10^{12}$ and plot four different values of $\epsilon$; in each case the percentage of volume of $S^\epsilon$ that is covered by $X^\epsilon$ quickly approaches $0$. Similarly, in Figure 16 (Center), if we fix $\epsilon = 1$ and plot four different values of $n$, in each case we have the same result. Finally in Figure 16 (Right) we plot a lower bound on number of samples necessary to cover $S^\epsilon$ by $X^\epsilon$ for four different values of $\epsilon$; in each case the number of samples necessary grows exponentially with the dimension.

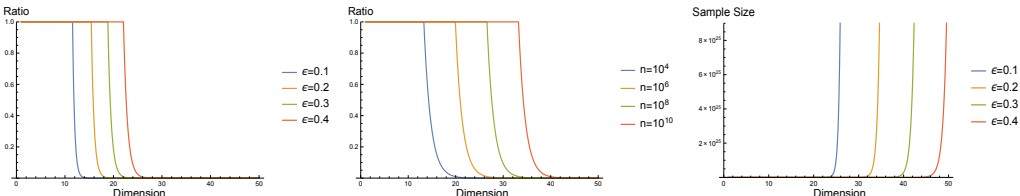

Figure 16: Three different perspectives on our upper bound in Equation 12. (Left, Center) In each case the percentage of $S^\epsilon$ covered by $X^\epsilon$ goes to $0$. (Right) The number of points necessary to cover $S^\epsilon$ by $X^\epsilon$ grows exponentially with the dimension.

## G   VORONOI DIAGRAMS AND DELAUNAY TRIANGULATIONS

Let $X \subset \mathbb{R}^d$ be a finite set of $n$ points. The *Voronoi diagram* of $X$, denoted $\mathrm{Vor}\,X$, under the metric $d(\cdot, \cdot)$ is a subdivision of $\mathbb{R}^d$ into $n$ cells where each cell is defined as

$$\mathrm{Vor}\,v = \{x \in \mathbb{R}^d : d(x, v) \leq d(x, u), \forall u \in X \backslash \{v\}\}. \tag{13}$$

In words, the Voronoi cell $\mathrm{Vor}\,v$ of $v \in X$ is the set of all points in $\mathbb{R}^d$ that are closer to $v$ than any other sample point $u \neq v$ in $X$. The Voronoi diagram is then defined as the set of all Voronoi cells, $\mathrm{Vor}\,X = \{\mathrm{Vor}\,v : v \in X\}$. When $d(\cdot, \cdot)$ is induced by the norm $\|\cdot\|_2$, the Voronoi cells are convex. See Figure 17.

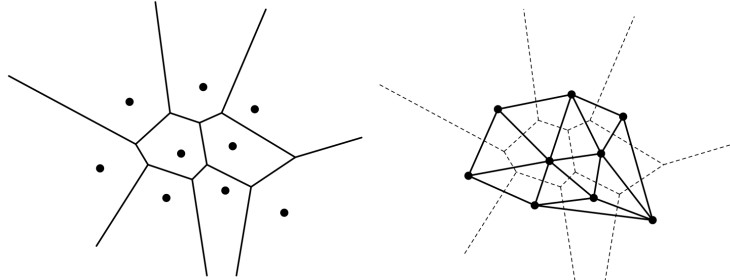

Figure 17: The Voronoi diagram of a set of points in $\mathbb{R}^2$ (left) and its dual the Delaunay triangulation (right).

The *Delaunay triangulation* of $X$, denoted $\mathrm{Del}\,X$ is a triangulation of the convex hull of $X$ into $d$-simplices. Every $d$-simplex $\tau \in \mathrm{Del}\,X$, as well as every lower-dimensional face of $\tau$, has the defining property that there exists an empty circumscribing ball $B$ such that the vertices of $\tau$ lie on the boundary of $B$ and the interior of $B$ is free from any points in $X$. See Figure 17. This *empty circumscribing ball* property of Delaunay triangulations implies many desirable properties that are useful in mesh generation (Cheng et al. (2012)) and manifold reconstruction (Edelsbrunner & Shah (1997)). The Delaunay triangulation of a point set always exists, but is not unique in general.

There exists a well known *duality* between the Voronoi diagram and the Delaunay triangulation of $X$. For every $j$-dimensional face $\sigma \in \mathrm{Vor}\,X$ there exist a dual $(d-j)$-dimensional simplex denoted $\sigma^* \in \mathrm{Del}\,X$ whose $d-j+1$ vertices are the $d-j+1$ vertices of $X$ whose Voronoi cells intersect at $\sigma$. In particular, every $d$-cell of $\mathrm{Vor}\,X$ is dual to the vertex of $\mathrm{Del}\,X$ that generates that cell, and every $(d-1)$-face of $\mathrm{Vor}\,X$ is dual to an edge of $\mathrm{Del}\,X$.

A nearest neighbor classifier $f_{nn}$ given a query point $q$ simply returns the class of the point in $X$ that generated the Voronoi cell in which $q$ lies. Thus the decision boundary of $f_{nn}$ is the union of $(d-1)$ and lower dimensional Voronoi faces. Furthermore, when $X$ is a dense sample of a manifold $\mathcal{M}$, the Voronoi cells are well known to be elongated in the directions normal to $\mathcal{M}$ Dey (2007). This fact underlies many of our results.

## H    VISUALIZATION OF DATASETS

In Figure 18 we provide visualizations of our two synthetic datasets, CIRCLES (left) and PLANES (right).

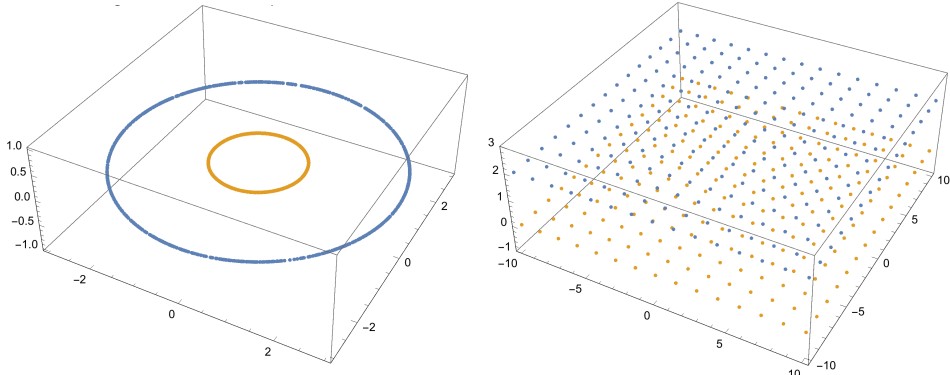

Figure 18: We create two synthetic datasets which allow us to perform controlled experiments on the affect of codimension on adversarial examples.

## I   VISUALIZATION OF DECISION BOUNDARIES

In Figure 19 we provide visualizations of the decision boundaries learned by (a-d) our fully connected network architecture with cross entropy loss for various optimization procedures and various training lengths, (e) our fully connected network architecture trained using the robust loss of Wong & Kolter (2018) for $\|\cdot\|_\infty$-perturbations, and (f) a nearest neighbor classifier for $\|\cdot\|_2$ on the training set. Specifically we train on the CIRCLES dataset, embedded in $\mathbb{R}^3$. The training set is entirely contained in the $xy$-plane. We then visualize cross sections of the decision boundary for various values of $z \in [-5, 5]$. We color points labeled as in the same class as the outer circle with the color blue and points labeled as in the same class as the inner circle as orange. Figure 19 shows the cross sections of the decision boundaries, averaged over 20 retrainings. The visualization shows how various optimization algorithms learn decision boundaries that extend into the normal directions where no data is provided.

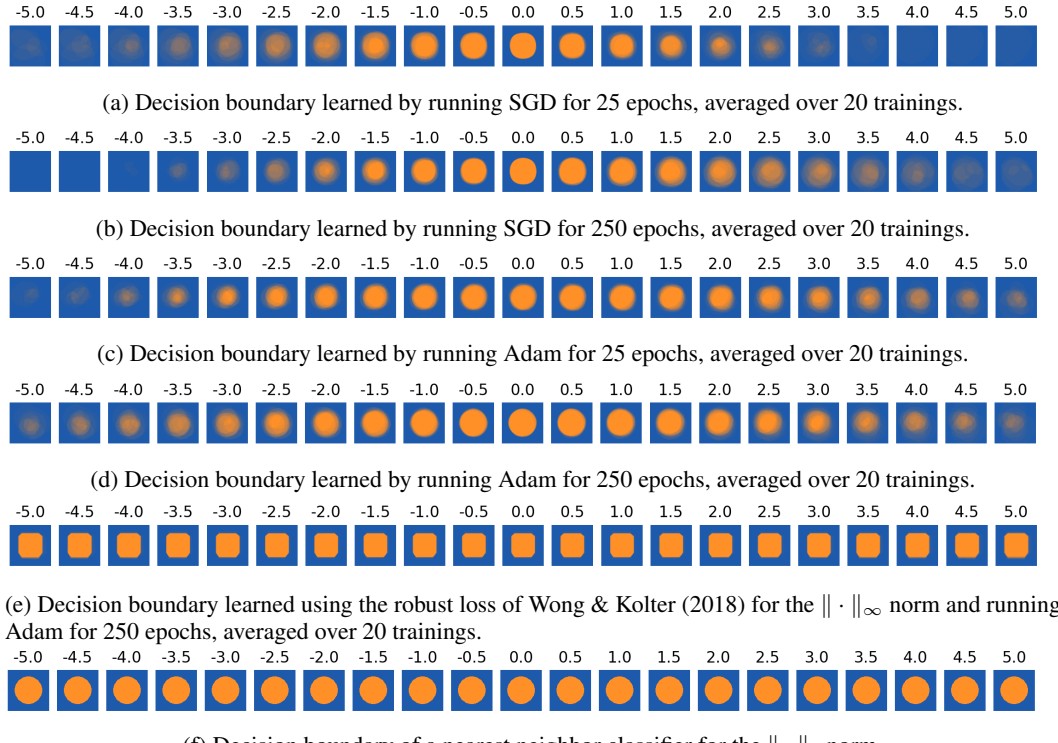

(a) Decision boundary learned by running SGD for 25 epochs, averaged over 20 trainings.

(b) Decision boundary learned by running SGD for 250 epochs, averaged over 20 trainings.

(c) Decision boundary learned by running Adam for 25 epochs, averaged over 20 trainings.

(d) Decision boundary learned by running Adam for 250 epochs, averaged over 20 trainings.

(e) Decision boundary learned using the robust loss of Wong & Kolter (2018) for the $\|\cdot\|_\infty$ norm and running Adam for 250 epochs, averaged over 20 trainings.

(f) Decision boundary of a nearest neighbor classifier for the $\|\cdot\|_2$ norm.

Figure 19: The training set lies entirely in the $xy$-plane, shown here at $z = 0$. We visualize cross sections of the decision boundary for $z \in [-5, 5]$ for various optimization algorithms training for different lengths of time. The results show how various optimization algorithm learn decision boundaries that extend into the normal directions in which there is no data provided. We average the decision boundary over 20 retrainings, so faded results indicated how frequently a point was labeled a specific class.

## J   COMPARING NEAREST NEIGHBOR ADVERSARIAL EXAMPLES WITH BIM ATTACK

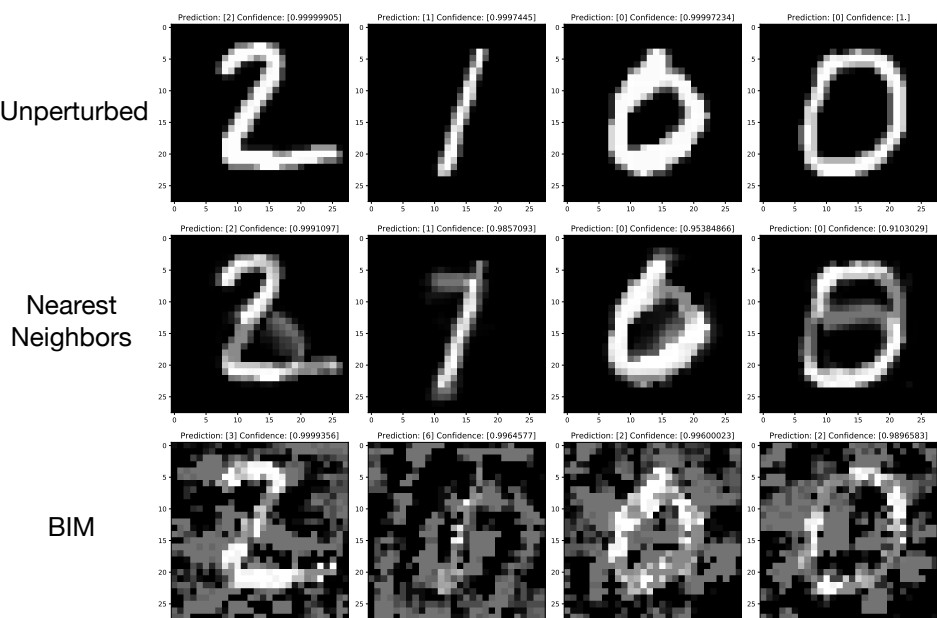

Figure 20: Comparison of adversarial examples for nearest neighbor with adversarial examples for Madry et al. (2018). The top row is the original data that the examples were generated from. Each figure is labelled with the predictions from robust neural network. We observe an immediate qualitative difference between the nearest neighbor examples and the BIM examples: the nearest neighbors ones are starting to look like numbers from a target class! In fact, we can reasonably argue that the classifications of the robust model that don't change represent as much of an error and being fooled by a standard adversarial example. For example the center right image would be classified as an 8 by most people, but the neural network is confident it is a 0. This provides evidence that nearest neighbors is doing a better job of the learning the *human* decision boundary between numbers.

