# OpenReview forum: "On the Geometry of Adversarial Examples"
_ICLR.cc/2019/Conference_

### Official Review · AnonReviewer2 · 2018-11-01
**An interesting paper on adversarial examples, but with certain concerns**

**Rating:** 6
**Confidence:** 4

**Review:**

This paper tried to analyze the high-dimensional geometry of adversarial examples from a geometric framework. The authors explained that there exists a tradeoff between being robust to different norms. They further proved that it is insufficient to learn robust decision boundaries by training against adversarial examples drawn from balls around the training set. Moreover, this paper showed that nearest neighbor classifiers do not suffer from this insufficiency.

In general, I think this paper is very interesting and enlightening. The authors analyzed the most robust boundary of norm 2 and norm infinity in different dimensions through a simple example and concluded that the single decision boundary cannot be robust in different norms. In addition, the author started from a special manifold and proposed a bound (ratio of two volumes) to prove the insufficiency of the traditional adversarial training methods and then extended to arbitrary manifold. It is good that this might provide a new way to evaluate the robustness of adversarial training method. However, I have some concerns: 1) Is it rigorous to define the bound by vol_X/vol_pi? In my opinion, the ratio of the volume of intersection (X^\del and \pi^\del) and vol \pi^\del may be more rigorous? 2) I don't know if such bound can be useful or easily applied in other work? In my opinion, it might be difficult, since the volume itself appears difficult to calculate.
I think the paper is a bit complicated or heavy in mathematics, and not easy to follow (though I believe I have well understood it). Some typos and minor issues are also listed as below.

Minor concerns:
1. At the end of the introduction, 3 attacking methods, FGSM, BIM, and PGD, should be given their full names and also citations are necessary.
2. Could you provide a specific example to illustrate the bound in Eq. (3), e.g. in the case of d=3, k=1.
3. In Page 7, “Figure 4 (left) shows that this expression approaches 1 as the codimension (d-k) of Pi increases.”  I think, the subfigure shows that the ratio approaches 1 when d and k are all increased.

---

> ### Author Response · Authors · 2018-11-09
> **Response to Reviewer 3**
>
> Thank you for your review. Please see our new post for common comments. Below we respond to your individual concerns.
>
> Re: Definition of volume ratios.
>
> You're correct that it would not be rigorous to define the bound using \vol X / \vol pi, as \vol X = 0, since the volume of a set of points is 0. We do not define the ratio in this way. Instead we define the ratio as \vol X^\del / \vol pi^\del.
>
> Your suggestion is indeed a rigorous way to compute this ratio. After some thought, your suggestion is equivalent to ours for this setting. In our setting, since the points are on the manifold pi, \vol X^\del is a subset of \vol pi^del, and so the intersection \vol X^\del \cap \vol pi^\del = \vol X^\del. It follows that (\vol X^\del \cap \vol pi^\del) / \vol pi^\del =  \vol X^\del / \vol pi^\del. We have added this argument to the paper.
>
> Re: Can volume bounds be applied to other work
>
> We had not considered applying this bound as a metric to evaluate the robustness of a specific adversarial training method. We think this is an interesting direction for future work. Toward this end, one may want to compute the intrinsic dimension of the data manifold. There has been prior work on estimating the dimension of manifolds from samples [1,2,3].
>
> In our summary we highlight that our key contribution is to identify the role of codimension on the pervasiveness of adversarial examples. In our paper, the primary purpose of our bounds is to prove that there exist classifiers that perform differently with respect to codimension. The specific values of these bounds show that this difference can, in theory, be exponentially sized. The primary practical use of our bounds is that they suggests that the transition from robust to non-robust is rapid; more like a phase transition, than a gradual shift.
>
> Re: Could you provide a specific example to illustrate the bound in Eq. (3), e.g. in the case of d=3, k=1.
>
> This is a good idea and we are currently considering the type of figure we might create to illustrate Equation 3. We note that we do provide an illustration in Figure 3 (right) to illustrate Equation 2, which is the more difficult step in deriving Equation 3. We updated the text to make the reference to that figure more prominent.
>
> Re: Subfigure in Figure 4
>
> Thank you for pointing out that we should have been more clear with our explanation. We can imagine two settings, one where we hold d fixed and increase k, and another where we hold k fixed and increase d. In the first setting, Figure 4 shows that lower dimensional problems are generally easier. This aligns well with results and intuition in the machine learning community. We are trying to draw attention to the second setting, that if we hold k fixed and increase d (and thus increase the codimension) the problem becomes more difficult.
>
> [1] Dimension Detection by Local Homology
> [2] Maximum Likelihood Estimation of Intrinsic Dimension
> [3] Estimating Local Intrinsic Dimension with k-Nearest Neighbor Graphs

---

### Official Review · AnonReviewer1 · 2018-11-02
**Synthetic examples and weak analysis of nearest neighbor classifier**

**Rating:** 3
**Confidence:** 4

**Review:**

This paper gives a theoretical analysis of adversarial examples, showing that (i) there exists a tradeoff between robustness in different norms, (ii) adversarial training is sample inefficient, and (iii) the nearest neighbor classifier can be robust under certain conditions. The biggest weakness of the paper is that theoretical analysis is done on a very synthetic dataset, whereas real datasets can hardly be conceived to exhibit similar properties. Furthermore, the authors do not give a bound on the probability that the sampling conditions for the robust nearest neighbor classifier (Theorem 1) will be satisfied, leading to potentially vacuous results.

While I certainly agree that theoretical analysis of the adversarial example phenomenon is challenging, there have been prior work on both analyzing the robustness of k-NN classifiers (Wang et al., 2018 - http://proceedings.mlr.press/v80/wang18c/wang18c.pdf) and on demonstrating the curse of dimensionality as a major contributing factor to adversarial examples (Shafahi et al., 2018 - https://arxiv.org/abs/1809.02104, concurrent submission to ICLR). I am very much in favor of the field moving in these directions, but I do not think this submission is demonstrating any meaningful progress.

Pros:
- Rigorous theoretical analysis.

Cons:
- Results are proven for particular settings rather than relying on realistic data distribution assumptions.
- Paper is poorly written. The authors use unnecessarily complicated jargon to explain simple concepts and the proofs are written to confuse the reader. This is especially a problem since the paper exceeds the suggested page limit of 8 pages.
- While it is certain that nearest neighbor classifiers are robust to adversarial examples, their application is limited to only very simple datasets. This makes the robustness result lacking in applicability.
- Weak experimental validation. The authors make repeat use of synthetic datasets and only validate their claim on MNIST as a real dataset.

---

> ### Author Response · Authors · 2018-11-09
> **Response to Reviewer 2**
>
> Thank you for your review. Please see our new post for common comments. Below we respond to your individual concerns.
>
> Re: Dimensionality in adversarial examples
>
> The review points to a concurrent submission to ICLR [1] as an example of prior work on the relationship between high dimensional input spaces and adversarial examples.
>
> Our paper is primarily concerned with the effect of codimension on robustness to adversarial examples. In this sense our paper is not about the curse of dimensionality, but rather of codimensionality. Thus, our work is complementary to [1]. We highlight the differences below.
>
> Shafahi et al. model each class as a probability density function defined on some domain. In contrast we define each class as a low dimensional manifold. In their model, our results consider the case where data lies on a measure 0 subset of the embedding space. Their results hold under the condition that “the class distribution is not overly concentrated” [1]. In their discussion the suggest extending their results to measure zero densities by considering a tube around the data. In our framework a robust classifier is one that accurately classifies the tube around the data. Our results show that learning an accurate classifier on the manifold is a fundamentally easier problem than learning an accurate classifier on the tube around the manifold, i.e. a robust classifier.
>
> We also note that the results in [1] are exclusive to spheres and cubes, which exhibit concentration of measure properties. Our paper cites [2] which also considered concentration properties of spheres and how they impact adversarial robustness.
>
> Re: Analysis on synthetic datasets
>
> One of our primary contributions is to exhibit a tradeoff in robustness under different norms. We show that this tradeoff exists in general by exhibiting a setting where the decision axes are not equal. Please see our response to R1 on how often this occurs. We have updated the paper to include this response. The remainder of our theoretical results apply to general manifolds and we have updated the text to make this clear.
>
> Re: Clarity
>
> We are sorry that you found the paper unclear. We made a concerted effort to make the paper readable, but it is clear that we can do better. We also note that this is an interdisciplinary paper, and one of the contributions of our work is to make a very difficult to access field more accessible to adversarial examples researchers. High dimensional geometry is highly counterintuitive. As a result, the field of computational geometry prioritizes clear rigorous formal proof and tools. We attempted to use the simplest tools that we could and we iterated multiple times to cut down on unnecessary definitions.  We also note that R1 commented positively on the clarity of our paper.
>
> We are highly committed to resolving any clarity issues with the paper and will be happy to incorporate any concrete suggestions that you have.
>
> Re: Applicability of nearest neighbor results
>
> As evidenced by the ICML paper that you provided (Wang et al. 2018), the robustness of nearest neighbor classifiers is an active and interesting research question. Wang et al. even provide a modification of the standard nearest neighbors algorithm that they show is more robust in practice.
>
> Our paper is not about nearest neighbor classifiers. As we described in the rebuttal to R1, it is important to understand why nearest neighbor classifiers are more robust in our setting. Nearest neighbors naturally handles high codimension settings because the Voronoi cells are elongated in the normal directions. We have modified the paper to make this point clear.
>
> Re: Concerns on experimental validation
>
> Our primary contribution is our theoretical results detailed in the summary above. Our experiments complement our theoretical results. Our synthetic training data is intended to explore the predictive power of our model of learners for real algorithms. Our results in Fig. 2 show that our theory for changing the norm predicts when real adversarial approaches fail. The CIRCLES and PLANES datasets show that real algorithms do, in fact, show this vulnerability to codimension. Our experiment on MNIST provides an example of a dataset with non-uniform sampling where nearest neighbor classifiers have fundamentally different performance than an adversarial training approach. We will update the paper to emphasize ways our experiments complement and support our other results. We have considered additional experiments that modify co-dimension for MNIST or the big-MNIST domain from [1] and would be happy to run them if requested.
>
> We would like to highlight the fact that we made careful effort to use state-of-the-art attacks and defenses and followed best practices when running the experiments (e.g. averaging over multiple retrainings).
>
> [1] Shafahi etal, Are adversarial examples inevitable?
> [2] Adversarial spheres
> [3] Adversarially robust generalization requires more data

---

### Official Review · AnonReviewer3 · 2018-11-03
**interesting work, but the theory is not very deep**

**Rating:** 5
**Confidence:** 3

**Review:**

This paper studies the geometry of adversarial examples under the assumption that dataset encountered in practice exhibit lower dimensional structure despite being embedded in very high dimensional input spaces. Under the proposed framework, the authors analyze several interesting phenomena and give theoretical results related to the necessary number of samples needed to achieves robustness. However, the theory in this paper is not very deep.

Pros:

The logic of this paper is very clear and easy to follow. Definitions and theories are illustrated with well-designed figures.

This paper shows the tradeoff between robustness under two norm and infinity norm for the case when the manifolds of two classes of data are concentric spheres.

When data are distributed on a hypercube in a k dimensional subspace, the authors show that balls with radius \delta centered at data samples only covers a small part of the ‘\delta neighborhood’ of the manifold.

General theoretical results on robustness and minimum training set to guarantee robustness are given for nearest neighbor classifiers and other classifiers.

Cons:

Most of the theoretical results in this paper are not very general. The tradeoff between robustness in different norms are only shown for concentric spheres; the ‘X^\epsilon is a poor model of \mathcal{M}^\epsilon’ section is only shown for hypercubes in low dimensional subspaces.

Section 5 is not very convincing. As is discussed later in the paper, although $X^\delta$ only covers a small part of \mathcal{M}^\delta, robustness can be achieved by using balls centered at samples with larger radius.

Most of the analysis is based on the assumption that samples are perfectly distributed to achieve the best possible robustness result. A more interesting case is probably when samples are generated on the manifold following some probabilistic distributions.

Theorems given in Section 6 are reasonable, but not very significant. It is not very surprising that nearest neighbor classifier is more robust than ‘x^\epsilon based’ algorithms, especially when the samples are perfectly distributed.

---

> ### Author Response · Authors · 2018-11-09
> **Response to Reviewer 1**
>
> Thank you for your review. Please see our new post for common comments. Below we respond to your individual concerns.
>
> Re: ‘X^\epsilon is a poor model of \mathcal{M}^\epsilon’ is only shown for hypercubes
>
> Thank you for pointing out that this section is unclear. The primary mathematical result in Section 5 is in Equation 4, which shows that this phenomena holds for general manifolds under additional conditions. To emphasize this, we have moved this result to the beginning of the section and under an explicit Theorem statement. We originally chose the order in the submission to first provide intuition to reader before introducing the more general result. We plan to keep that body of text but to more clearly contextualize it as an intuitive walkthrough in a special case. We invite additional suggestions for improvements.
>
> Re: Robustness tradeoff only shown for concentric spheres
>
> It is a good question to ask how often does the L2 decision axis differ from the Linf decision axis. We believe that it is the common case that the L2 decision axis differs from the Linf decision axis, and that this phenomena explains recent results on adversarial robustness under different norms [1]. The result extends easily to, e.g., two concentric cylinders. Consider a two-dimensional axis-aligned cross section; in this cross section, the fact that the optimal decision boundaries differ is a corollary of our result. A similar argument works for intertwined tori. We have updated the paper to include a discussion of this intuition and will update the paper with any additional formal results we develop. We also note that our proof uses the spheres in a proof by construction. We made this choice for readability. This is common in theory on adversarial examples. [2] uses theoretical results about linear classifiers and simple data distributions to provide insight into robustness for adversarial examples.
>
> Additionally, our work provides a new source of problems, motivated by adversarial examples, for the computational geometry community. The CG literature has been motivated by reconstructing manifolds, for which the decision axes under the L2 norm is all that is needed. The CG community has not explored the geometry of max-margin decision boundaries under norms other than L2.
>
> Re: Using larger balls to achieve robustness.
>
> We are unsure of the meaning of this crique. In the statement “As is discussed later in the paper … robustness can be achieved by using balls centered at samples with larger radius”, can you clarify which discussion you are referencing?
>
> We consider ball-based learners. In Section 5 we show that the ratio of the volume of the union of balls around the samples to the volume of the tubular neighborhood approaches 0 as the codimension increases. This causes problems for ball-based learners because the measure captured by the balls is 0 in high codimensions. A natural way to remedy this is to use larger balls. However there are limits to how large the balls can be made before they begin to intersect the balls from samples on a different class manifold. Theorem 1 shows that even when ball-based learners use the largest possible balls, they require many more samples to achieve the same amount of robustness.
>
> Re: Proposed extension to non-uniform distributions on data manifolds.
>
> We are very interested in the direction you suggested. In future work we intend to combine techniques from the statistics literature with our framework, including sampling according to some distribution on a manifold, to understand the more difficult setting and prove more realistic guarantees for learning algorithms.
>
> In this paper our first goal is to bridge two disparate communities, leveraging the techniques from the manifold reconstruction literature to provide a different perspective and new tools for the problem of adversarial examples. We wish to understand the simplest version of the problem, which, as shown in our paper, already makes several issues clear, such as the affect of codimension on adversarial robustness.
>
> Re: Significance of robustness of nearest neighbors versus  ‘x^\epsilon based’ algorithms.
>
> We apologize for not making the point of the results in Section 6 clear. The importance of Theorem 1 is to show that different classification algorithms have different sampling requirements with respect to robustness. In particular nearest neighbor classifiers require fewer samples to achieve the same level of robustness for a fixed codimension. The ball-based learner is a theoretical model of the adversarial training used in state-of-the-art defenses for adversarial examples [3,4]. We have updated this section to make the importance of our results more clear.
>
> [1] Towards the first adversarially robust neural network model on MNIST.
> [2] Adversarially robust generalization requires more data. NIPS
> [3] Explaining and harnessing adversarial examples. ICLR
> [4] Towards deep learning models resistant to adversarial attacks. ICLR

---

### Author Response · Authors · 2018-11-09
**Common Response to All Reviewers**

We would like to thank all the reviewers for their helpful comments. To avoid repetition, we present general comments here and individual comments below.

In reading the reviews we realized that our introduction was unclear with respect to the contributions of the paper. We have restructured the introduction to appropriately highlight our contributions. The primary contributions of the paper are as follows. First we introduce a geometric framework, where we model classes of data as lying on distinct manifolds. Second we use this framework to show that there exists a tradeoff in robustness under different norms. Third, we show that in theory high codimension plays a role in vulnerability to adversarial examples. Vulnerability to adversarial examples is often attributed to high dimensional input spaces. To our knowledge this is the first work that investigates the role codimension plays in adversarial examples. We give theoretical results that show that even under ideal sampling conditions, state of the art methods, like adversarial training, fail in simple settings. Interestingly we find that different classification algorithms are less sensitive to changes in codimension. In preliminary experiments on synthetic data and on MNIST we provide empirical evidence to support this point.

Regarding the related work of Wang et al. (ICML 2018) on kNN. We were unaware of the work of Wang et al. and we would like to thank the R2 for bringing this important related work to our attention. In developing the paper we only turned to nearest neighbor as an example of a classification algorithm that is robust to high-codimension. We apologize for the lack of clarity. The work of Wang et.al. is related and we have updated the paper to appropriately contextualize our results with respect to this work. Specifically we have added the following passage to the related work.

“Wang et al. (2018) explore the robustness of k-nearest neighbor classifiers to adversarial examples. In the setting where the Bayes optimal classifier is uncertain about the true label of each point, they show that k-nearest neighbors is not robust if k is a small constant. They also show that if k is asymptotically large, then k-nearest neighbors is robust. Using our geometric framework we show a complementary result: in the setting where each point is certain of its label, 1-nearest neighbors is robust to adversarial examples.”

Approaching the problem from a geometric perspective, we reach the complementary result that 1-nearest neighbors is robust in the setting where each sample is certain of its true label.

---

### Public Comment · ~Tianhang_Zheng1 · 2018-11-27
**Very interesting results**

Very interesting results. Although the theory is only proved for some simple topologies, still got a lot of insights from this paper.

Just one small question: how did you implement BIM on KNN classifiers? Maybe it is already introduced in the paper, but I did not find it.

---

> ### Author Response · Authors · 2018-11-28
> **Response**
>
> Hi Tianhang. Thank you, we're glad you found the paper enlightening. Could you clarify to which result you're referring to specifically?

---

> > ### Public Comment · ~Tianhang_Zheng1 · 2018-11-28
> > **Clarification**
> >
> > Just a minor question: what I refer to is the result in figure 6. BIM attack is tested on K-NN (1-NN). I was wondering how did you implement BIM on K-NN? Is K-NN differentiable?

---

> > > ### Author Response · Authors · 2018-11-28
> > > **Response**
> > >
> > > In Figure 6 Left and Center, we compute adversarial examples using BIM for the natural (left) and robust (center) models and classify those adversarial examples using both the the model and a nearest neighbor classifier. In the paper we state "At eps = 0.5, nearest neighbors maintains accuracy of 78% to adversarial perturbations that cause the accuracy of the robust model to drop to 0%". This is what we expect if the adversarial perturbations are in directions nearly normal to the data distribution where nearest neighbors naturally excels due to the geometric properties of its decision boundary.
> > >
> > > However NN has its own failure modes. In Figure 6 Right, we consider a custom iterative attack on a nearest neighbor classifier, as described in the second paragraph of Section 7.2. In this case the robust model is more successful at classifying the adversarial examples generated for nearest neighbors using this custom attack, implying that their failure modes are distinct. In Appendix J, Figure 20 we provide a qualitative comparison of the adversarial examples generated for both the robust model (using BIM) and nearest neighbors (using our custom attack). Figure 20 shows immediate qualitative differences between the two.
> > >
> > > We hope this answers your question. Let us know if you have any others.

---

> > > > ### Public Comment · ~Tianhang_Zheng1 · 2018-11-28
> > > > **Thanks for your reply**
> > > >
> > > > Got it. So the BIM adversarial samples are crafted from the natural model and Madry's robust model (For NN, BIM is like a black-box attack).
> > > >
> > > > Thanks for your reply!

---

### Public Comment · (anonymous) · 2018-12-01
**Missing discussion?**

In paper [1], they found that adversarial examples escape to submanifold/subspace of higher intrinsic dimensionality, which seems equivalent to the findings proposed here. Given a fixed embedding/representation dimension, the lower the intrinsic dimension (of the underlying manifold), the higher the codimension (higher vulnerability), and also the easier escape to "higher" intrinsic dimensionality. Suppose embedding dimension d=10, intrinsic dimension k=2 vs k=5:
1) findings in this paper: codimension=10 - 2 (8, higher vulnerability) vs 10 - 5 (5);
2) findings in [1]: the intrinsic dimensionality of adversarial submanifold should have k>2 (also indicating higher vulnerability, as it is much easier to escape to k>2 than k>5) vs k>5. I am wondering, what makes codimension based analysis different to intrinsic dimensionality based analysis, as for a given dataset, its embedding dimension is fixed. Sorry if I misunderstood the idea.

[1] Characterizing Adversarial Subspaces Using Local Intrinsic Dimensionality. ICLR 2018

---

> ### Author Response · Authors · 2018-12-06
> **Response**
>
> Hi,
>
> Apologies for the slow response. The authors of [1] propose LID as a measure of the “intrinsic dimensionality” of a point with respect to a given dataset. The authors show that adversarial examples tend to exhibit higher LID than unperturbed examples and explore using LID features as a detector for adversarial examples. We note however that, as shown in Figure 1 of [1], LID may be larger than the embedding dimension (LID = 4.36 in a 2D embedding space). Thus LID is not easily interpretable as the dimension of a subspace as claimed. In our language, points in the normal directions off of the data manifold would exhibit higher LID. However higher LID is not necessarily indicative of an adversarial example. Whether or not a point off the data manifold is an adversarial example is dependent upon the decision boundary of the classifier. For example, many points off the data manifold in our examples may exhibit high LID, but the decision axis still classifies such points correctly. For the optimal decision boundary such points are not adversarial examples. Furthermore the local neighborhood near an adversarial example need not have the geometry of an affine subspace, and can exhibit more complex geometry depending on the decision boundary.
>
> We draw attention to codimension as a key source of the pervasiveness of adversarial examples. Codimension is an exact characterization of every possible direction off of the data manifold, and is always equal to d - k. Unlike the LID, which characterizes the local dimensionality of a single point with respect to a data set, the normal space is a linear subspace of dimension d - k which captures all of the normal directions off of the manifold. When the codimension is high, there are many directions off of the data manifold in which to construct adversarial examples. We show empirically that in high codimension settings, standard optimization procedures and adversarial training have difficulty learning a decision boundary that is far away from the data manifold in every normal direction. Thus we conclude that high codimension increases vulnerability.

---

### Public Comment · ~Emin_Orhan1 · 2018-12-05
**connection to Wang et al. (ICML, 2018)**

I'm not sure if the added text comparing with Wang et al. (ICML, 2018) captures the connection between the two works accurately enough. The main message of Wang et al. (2018), as I read it, is that there's a fundamental difference between adversarial robustness characteristics of k-nn classifiers with small k and those with large k. It seems to me that Theorems 5-6 in the current paper can be significantly improved for a k-nn with large k as well (instead of the current k=1 case). In fact, for large k, it seems to me that delta can be made arbitrarily large. I would encourage the authors to consider this case in a future revision.

I think overall this paper provides useful insights. I particularly appreciate the results on the nearest neighbor classifiers. I think the robustness of nearest neighbor type models is underappreciated in the current literature on adversarial examples. Finally, I would like to point out a few papers that came out recently empirically demonstrating the superior adversarial robustness properties of these kinds of models. It may be useful for the authors to know about these more empirically motivated papers (disclosure: I'm the author of the last one listed below):

1. Zhao J, Cho K (2018) Retrieval-augmented convolutional neural networks for improved robustness against adversarial examples. arXiv:1802.09502.

2. Papernot N, McDaniel P (2018) Deep k-nearest neighbors: Towards confident, interpretable and robust deep learning. arXiv:1803.04765.

3. Orhan AE (2018) A simple cache model for image recognition. NeurIPS 2018. arxiv:1805.08709.

---

> ### Author Response · Authors · 2018-12-06
> **Response**
>
> Hi Emin,
>
> We’re glad you found our paper insightful. In particular we’d like to thank you for bringing additional references to our attention.
>
> > Regarding potential improvements to our results on k-nn.
>
> This is actually not true in our mathematical model. The reason it is not true is because we place no condition forbidding “oversampling”. While a delta-cover requires that every point on the data manifold has a sample within a distance delta, there is no condition that forbids a point from having arbitrarily many such samples. In particular we can construct examples in which the nearest sample is on the correct class manifold while the next k-1 samples are on a different class manifold. The precise number of points that are guaranteed to be on the correct class manifold will vary as we move throughout the tubular neighborhood, but we can always construct a sampling situation in which the majority of the k samples are on the wrong class manifold, for sufficiently large k. These configurations are unlikely in practice, and it may be reasonable to impose a condition such as “no two samples are closer than some distance alpha” to prevent oversampling. With this additional condition it may be possible to prove something as you’ve described. Alternatively one could consider a statistical setting where points are sampled from a probability distribution with a specific support. As we’ve discussed in one of our rebuttals, we’re very interested in the direction of point sets sampled from class manifolds according to some probability distribution. However we feel that the lovely work of Wang et al has already well explored k-nn classifiers in this setting.
>
> Furthermore the reason we considered specifically 1-nearest neighbors is because the decision boundary induced by 1-nearest neighbors is comprised of Voronoi facets, and it is well known that the Voronoi cells are elongated in the directions orthogonal to the data manifold for dense samples. Thus 1-nearest neighbors is an example of a classification algorithm that naturally accounts for the high codimension of the data manifold, which we have argued is a key source of the pervasiveness of adversarial examples for naturally trained and adversarially trained deep networks.

---

### Meta-Review · Area_Chair1 · 2018-12-18
**Interesting work, but restrictive analysis**

**Confidence:** 4
**Recommendation:** Reject

**Metareview:**

The paper gives a theoretical analysis highlighting the role of codimension on the pervasiveness of adversarial examples. The paper demonstrates that a single decision boundary cannot be robust in different norms. They further proved that it is insufficient to learn robust decision boundaries by training against adversarial examples drawn from balls around the training set.

The main concern with the paper is that most of the theoretical results might have a very restrictive scope and the writing is difficult to follow.

The authors expressed concerns about a review not being very constructive. In a nutshell, the review in question points out that the theory might be too restrictive, that the experimental section is not very strong, that there are other works on related topics, and that the writing of the paper could be improved. While I understand the disappointing of the authors, the main points here appear to be consistent with the other reviews, which also mention that the theoretical results in this paper are not very general, that the writing is a bit complicated or heavy in mathematics, and not easy to follow, or that it is not clear if the bounds can be useful or easily applied in other work.

One reviewer rates the paper marginally above the acceptance threshold, while two other reviewers rate the paper below the acceptance threshold.